# High-throughput proteome integral solubility alteration assay for low cell input using One-Tip
Maico Lechner [1], Pierre Sabatier [1,2] ✉ & Jesper V. Olsen [1] ✉

Mass spectrometry-based versions of the cellular thermal shift assay (CETSA), like proteome integral solubility alteration (PISA), enable simultaneous monitoring of thousands of proteins for drug-target engagement. These methods are constrained in throughput and scalability, while the sample requirement limits the applicability to widely available material. Here, we combine PISA with the One-Tip method to simplify and streamline sample preparation. Using the mass spectrometry-compatible *n*-Dodecyl-β-D-Maltoside (DDM) non-ionic detergent for cell lysis in PISA sample preparation enables direct transfer to One-Tip with decreasing cell requirements down to 200 cells per μL. One-Tip provides similar depth and higher reproducibility, with lower material and solvent usage and a faster proteolytic digestion compared to a conventional sample cleaning and digestion protocol, making it a cost-effective, fast, and user-friendly option. To demonstrate its scalability, we applied One-Tip-PISA in a 96-well plate format, profiling a kinase inhibitor panel, allowing cell treatment to injection within 12 h, enhancing workflow efficiency and accessibility for a wide range of laboratories.

Methodologies based on the cellular thermal shift assay (CETSA) are routinely used to study drug-target engagement and mechanisms of action[1]. It is based on the general phenomenon that proteins denature and become insoluble with increasing temperature, but their thermal stability changes by interactions with small molecules[2]. Particularly, mass spectrometry (MS)-based proteomics variants such as thermal proteome profiling (TPP) or its higher-throughput version, proteome integral solubility alteration (PISA) enable global analysis of thousands of proteins[3,4]. These methods have been widely used to study the effects of drug treatments in cell lines or animal tissues[5,6]. However, they are still limited in throughput, and the multi-step sample preparation workflow makes them hard to automate and usually requires a high sample input amount. While recent developments have addressed either of these issues, no method has currently managed to address all of them simultaneously[5,7]. In addition, although widely applicable, current PISA workflows face challenges as they are restricted to soluble proteins. This limits their efficiency and accessibility, especially for membrane receptor proteins, which are prominent targets in medical and pharmaceutical research, where sample availability is often constrained. To profile membrane protein drug targets, it is necessary to use a detergent during cell extraction. For instance, the use of non-MS-compatible detergents for cell lysis, like NP40 which necessitates additional steps, such as protein precipitation, filtration, or protein aggregation capture (PAC), to

remove the detergent prior to LC-MS analysis. This increases the risk of protein loss and variability due to extra sample handling steps. These methods, while robust and capable of achieving high proteome depth, also demand long proteolytic digestion times, and the use of other consumables such as cartridges and magnetic beads, which add to the expense and complexity of the workflow.

Here, we introduce an optimized workflow that combines the One-Tip method with PISA analysis. This novel integration leverages a streamlined sample preparation in the Evotip, enabling direct protein digestion, desalting, and concentration of peptides within the C18 resin of the tip, minimizing sample handling and reducing the need for large input amounts. Moreover, using 0.2% *n*-Dodecyl-β-D-Maltoside (DDM), a mass spectrometry-compatible detergent, in the freeze-thaw lysis step removes the need for buffer exchange prior to proteolytic digestion, thus streamlining the method, reducing preparation time and variability[8]. Furthermore, only microliters of acetonitrile, isopropanol, formic acid, and Triethylammonium bicarbonate (TEAB) are needed for activating the EvoTips, as well as for the proteolytic digestion master mix with Lys-C and trypsin, reducing the amount of solvent and hazardous chemicals required for sample preparation.

To validate the sensitivity and reliability of One-Tip-PISA, we analyzed staurosporine (STS)-treated HeLa cells at varying cell inputs, using narrow-window data-independent acquisition (nDIA) on the Orbitrap Astral mass

[1]Department of Cellular and Molecular Medicine, Novo Nordisk Foundation Center for Protein Research, Faculty of Health and Medical Sciences, University of Copenhagen, Copenhagen, Denmark. [2]Cardio-Thoracic Translational Medicine (CTTM) Lab, Department of Surgical Sciences, Uppsala University, Uppsala, Sweden. ✉e-mail: pierre.sabatier@uu.se; jesper.olsen@cpr.ku.dk

spectrometer[9]. This approach allowed us to identify kinase targets effectively, with strong correlations to known STS targets, even at cell inputs as low as 200 cells/μL. The One-Tip-PISA method showed comparable performance in protein depth and superior reproducibility compared to existing PAC protocols for low sample amounts, while decreasing the processing time and cost significantly. Furthermore, the method's compatibility with 96-well plate formats enabled high-throughput processing, allowing for the analysis of a full plate in a single day of HeLa cells treated with a panel of kinase inhibitors targeting the EGF receptor signaling network, as well as several tyrosine kinase and non-kinase inhibitors applied to SCC25 cells, and with the application of fast LC-gradients sample data can be acquired within another day[10].

Through this study, we aimed to establish One-Tip-PISA as a label-free, streamlined, scalable, and low-input approach to PISA analysis. This method not only broadens PISA's applicability across various research fields but also maintains high-quality proteome coverage, aligning with advances in liquid chromatography tandem mass spectrometry (LC-MS/MS) that enable comprehensive proteomic analysis from minimal sample quantities.

## Results and discussion
### Impact of DDM concentration on cell lysis in PISA
In a PISA assay, the use of detergent during cell lysis is critical for improving protein recovery and particularly membrane-bound proteins. Many detergents are incompatible with MS workflows, but DDM provides an advantage by allowing efficient and streamlined sample processing without the need for detergent removal by multi-step protein digestion strategies like PAC and desalting, reducing the overall pipetting steps. The One-Tip workflow relies on 0.2% DDM, and while it is also a compatible detergent for PISA analysis, it was only tested at a concentration of 1%[8]. Thus, we first evaluated the effect of DDM concentration in a combined One-Tip-PISA analysis of intact cells. HeLa cells were treated with 10 μM STS or DMSO control and processed via the PISA workflow coupled to One-Tip (Fig. 1A). STS is a broad-spectrum, ATP-competitive protein kinase inhibitor often used to evaluate CETSA-based proteomics methodologies. Different concentrations of DDM (1.0%, 0.6%, and 0.2%) were tested for cell lysis after heat treatment, alongside a PBS control. After centrifugation, 5 μL of the supernatant was directly transferred to One-Tip-ready Evotips for protein digestion and LC-MS/MS analysis (Fig. 1A). As shown in Fig. 1B, C, the use of DDM significantly increased the number of recovered proteins and peptides compared to the PBS control. For example, the comparison between 0.2% DDM and the PBS control, both treated with 10 μM STS, resulted in $p$-values of 0.00005 for protein numbers and 0.00004 for peptide numbers. These results highlight the critical role of DDM as a detergent in enhancing protein recovery after the PISA assay. Although DDM improved protein and peptide recovery, the differences between 1.0%, 0.6%, and 0.2% were less pronounced, suggesting that beyond a certain threshold, increasing DDM concentration does not proportionally enhance recovery.

Additionally, DDM not only increased the number of identified proteins and peptides but also improved sample reproducibility during cell lysis compared to the PBS control (Fig. 2A). DDM-lysed samples also showed a lower median variance compared to the PBS-lysed samples, except for the 10 μM STS treatment, highlighting the enhanced consistency provided by DDM during the lysis process. Coefficient of variation (CV) values were calculated based on median-normalized non-log transformed data following the recommendations of Brenes et al[11]. We then analyzed the subcellular localization of protein quantified in the different DDM concentrations using the deep learning based eukaryotic subcellular localization predictor, DeepLoc 2.0[12]. The heatmap shows that across all subcellular fractions, DDM treatments (1.0%, 0.6%, and 0.2%) consistently result in higher protein extraction efficiency than PBS. In particular, the cell membrane, endoplasmic reticulum, lysosome, vacuole, and mitochondria showed significantly higher protein recovery with DDM than with PBS (Fig. 2B), highlighted by the results of the one-way analysis of variance (ANOVA), which evaluates differences in treatment groups (Supplementary Table 1). The one-way ANOVA results confirmed highly significant differences

($p < 0.0001$) across the median normalized log2 signal intensities in all five examined subcellular fractions, with the strongest effects observed in the endoplasmic reticulum ($p = 5.6 \times 10^{-14}$, $F = 1134$) and mitochondria ($p = 1.8 \times 10^{-12}$, $F = 599$), where the addition of DDM substantially increased signal intensity. Since One-Tip relies on 0.2% DDM for sample preparation, and increasing the concentration in PISA to up to 1% did not substantially improve the analysis, we opted to use 0.2% DDM for the rest of the study.

We further compared the 0.2% DDM condition to the PBS control (Fig. 2C), showing that the absence of DDM results in fewer identified kinases. In the 0.2% DDM condition, 22 kinases were identified, whereas only 15 passed the significance threshold (FC > 1 or < −1 and adjusted-$p$-value < 0.05) in the PBS condition. Additionally, 276 proteins in the PBS-treated samples passed the threshold, compared to 31 proteins in the 0.2% DDM-treated samples. Particularly, members of the Protein Kinase C family (*PRKCE*, *PRKCD*), were found uniquely in the DDM-containing lysis conditions, while they were notably absent in the fraction without DDM[13]. This trend was also observed for 1.0% and 0.6% DDM, which showed comparable results in identifying STS kinase targets (Supplementary Fig. 3). This supports that DDM effectively preserves the structure and enriches these membrane-bound kinases. Considering only membrane-associated protein kinases, a cluster heatmap shows that most protein kinases are not identified in the PBS fraction and that a 1.0% DDM concentration might be too high, as some kinase intensity starts to decrease compared to the other DDM fractions (Supplementary Fig. 1). Moreover, several known membrane-associated kinases have been found exclusively in the DDM-containing lysis fraction, such as *EGFR*, *AXL*, *FYN*, *YES1*, *SRC*, and *ROR2* (Fig. 2D)[14–17]. Tyrosine kinases like *EGFR* and *SRC* were significantly destabilized by STS treatment in HeLa cells, whereas other membrane protein kinases like *AXL*, *PDGFRB*, and *AAK1* are stabilized in the 0.2% DDM fraction. Lastly, all three DDM concentrations (1.0%, 0.6%, and 0.2%) lead to significantly more detected kinases compared to the PBS condition (Fig. 2E). The most striking difference is seen with the 0.6% DDM condition, which resulted in the detection of 62 unique kinases, compared to only 4 kinases detected in the PBS control. Consequently, the correlation was low between DDM and PBS control but high between DDM concentrations, further underscoring the advantage of DDM over PBS in kinase detection (Supplementary Fig. 4). Thus, we used 0.2% DDM for this study as it is also the concentration used in One-Tip preparation. Since membrane proteins are of broad interest in proteomics and have been investigated using thermal solubility-altering methods such as TPP or PISA, we compared our dataset to two published datasets from Batth et al. (0.2% NP40) and Reinhard et al. (0.4% NP40)[5,8]. Using 0.2% DDM, we detected 367 membrane proteins, compared to 167 in Batth et al. and 57 in Reinhard et al. (Supplementary Fig. 5). In terms of log2 summed intensities, both our dataset and that of Batth et al. display similar distributions between membrane and non-membrane proteins (4.7% vs 3.3%, respectively). In contrast, the Reinhard et al. dataset shows only 0.1% membrane protein intensity, likely due to the use of an older-generation mass spectrometer and differing sample preparation protocols (Batth et al.: Orbitrap Exploris 480; Reinhard et al.: Q Exactive Orbitrap).

### Evaluation of the cell concentration limit for One-Tip-PISA analysis
While LC-MS/MS analysis of tryptic digests of whole cell extracts from 500-1000 cells consistently provides very high coverage in proteomics nowadays (>9000 proteins quantified)[18], PISA in intact cells usually requires a significantly higher number of starting cells due to the inefficient lysis. Thus, we aimed to test the cell concentration required to generate reliable PISA analysis, using One-Tip-PISA in combination with the Orbitrap Astral MS. For this experiment, we treated HeLa cells with either vehicle (DMSO), 5 μM or 10 μM STS and prepared cell dilutions with the following concentrations: 800 cells/μL, 600 cells/μL, 400 cells/μL, and 200 cells/μL, and used 12.5 μL as input corresponding to 10,000, 7500, 5000, and 2500 cells per temperature point, respectively. We first measured the protein concentration after cell lysis using a BCA assay which indicated levels below the detection limit at

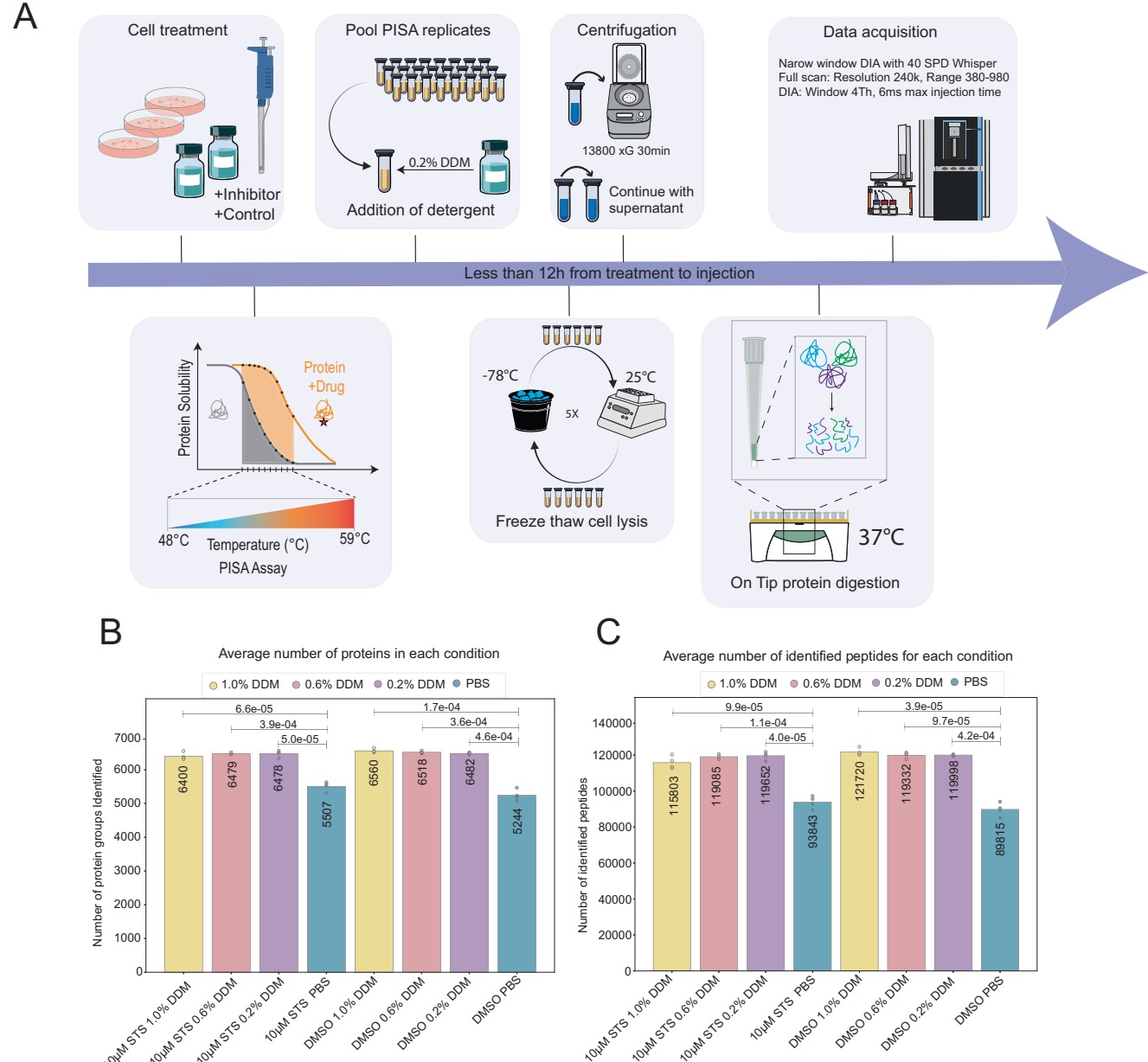

**Fig. 1 | One-Tip-PISA workflow and evaluation of the DDM concentration.**
**A** Visual abstract of the One-Tip-PISA method from sample preparation to data
acquisition. **B** Bar charts show the average number of identified proteins and

peptides (**C**), as well as *p*-values from a Welch's *t*-test across varying concentrations
of DDM detergent. Individual replicates are represented as dots (*n* = 4 for all con-
ditions except for 10 μM STS with 0.6% DDM, where *n* = 3).

200 cells/μL, highlighting the poor freeze-thaw lysis efficiency and protein
recovery, as expected (Supplementary Fig. 2). Orbitrap Astral analysis
showed consistent protein identification across treatments at 400 cells/μL
and 800 cells/μL (~6200–6300 protein groups; Supplementary Fig. 6A, B).
However, at 200 cells/μL, protein identification dropped significantly, with
peptide identification showing an even greater decline of over 25k fewer
peptides than at 400 cells/μL in some cases. To further evaluate the sensitivity
of the One-Tip-PISA method, we compared the results from 800 cells/μL to
those from 200 cells/μL, both treated with 5 μM STS (Fig. 3A). As shown in
Fig. 3A, at 800 cells/μL, we observed a clear engagement of the broad-
spectrum kinase inhibitor STS with kinases such as *AURKA* and *PAK3*.
These kinases are also affected when only 200 cells/μL are used, but the log2
FC and −log10 adjusted *p*-values are substantially smaller, and the number
of statistically significant kinases is greatly reduced at this lower con-
centration. Although the distributions of the log2 FC across the four cell
dilutions[19,20] (Fig. 3B) remain consistent between 800 and 400 cells/μL, they

increase at 200 cells/μL. The differences between the FC distribution at each
cell concentration compared to the distribution of 200 cells/μL was sig-
nificant, indicating an increase in variation of the log2 FC values and thus
noisier data (Fig. 3B). The Spearman correlation between 800 cells/μL, and
600 and 400 cells/μL are 0.81 and 0.84, respectively, indicating consistent
results between those conditions (Fig. 3C). However, at 200 cells/μL, the
correlations against other cell concentrations range between −0.62 and
−0.21, suggesting that the One-Tip-PISA method reaches its sensitivity limit
at this lower cell dilution. Principal Component Analysis (PCA) comparing
the different treatment conditions across four cell concentrations: 800, 600,
400, and 200 cells/μL highlighted two distinct clusters separated by the first
principal component (PC1) on the *x*-axis (Fig. 3D). The samples belonging
to the 200 cells/μL condition separate from the 800, 600, and 400 cells/μL
conditions, which form a separate cluster. PC1 captures the majority of
variance between these groups, implying that the variance between higher
cell concentrations and the 200 cells/μL condition is the dominant source of

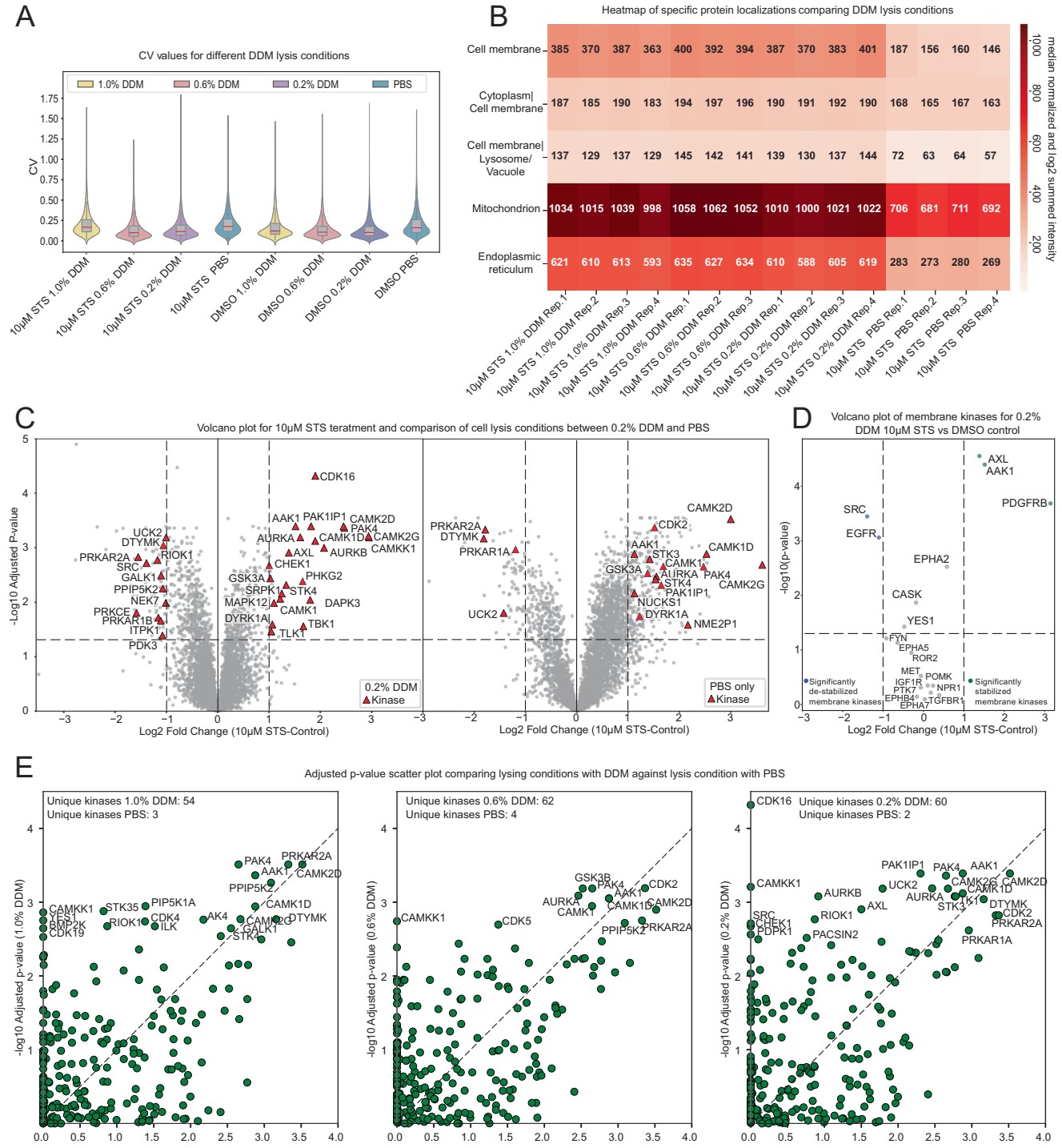

**Fig. 2 | Comparison between DDM and PBS cell lysis after PISA assay. A** Violin plots show the CV between different DDM concentrations and PBS control; median, first, and third quartiles are highlighted with a boxplot inside each violin. **B** Heatmap of median-normalized, log2-transformed summed protein signal intensities across DDM variations and PBS control, segmented by cellular subregions. **C** Volcano plots comparing 0.2% DDM to PBS for cell lysis, plotting log2 fold change (log2FC) against BH-adjusted −log10 $p$-values for 10 μM STS vs DMSO control; significant kinases (adjusted $p < 0.05$ and log2FC > 1 or < −1) are marked in red. **D** Volcano plot after 10 μM STS treatment of HeLa cells against DMSO control for 0.2% DDM cell lysis comparing FC and $p$-value changes of membrane protein kinases. Highlighted in green are significantly stabilized membrane kinases, whereas highlighted in blue are significantly destabilized membrane kinases. **E** Plot of BH-adjusted $p$-values for DDM concentration variations compared to PBS; kinases with adjusted −log10 $p$-values > 2.5 are labeled, along with unique identifications found exclusively in PBS or specific DDM samples.

variation in the dataset, further highlighting that the analysis is much less reliable at 200 cells/μL. While STS is a broad range kinase inhibitor that has been used in HeLa cells several times[19,20]. We also tested One-Tip-PISA, with different cell lines and a non-kinase inhibitor, to highlight the versatility of the method. For this, epithelial-like SCC25 cells were treated with the allosteric SHP-2 inhibitor RMC4550 in 1 μM and 5 μM concentrations, respectively. The results are depicted in Fig. 3E. The $p$-value and FC of the direct target, tyrosine phosphatase *PTPN11* (SHP-2), show a clear concentration-dependent effect, indicating that One-Tip-PISA can also be applied to non-kinase inhibitors.

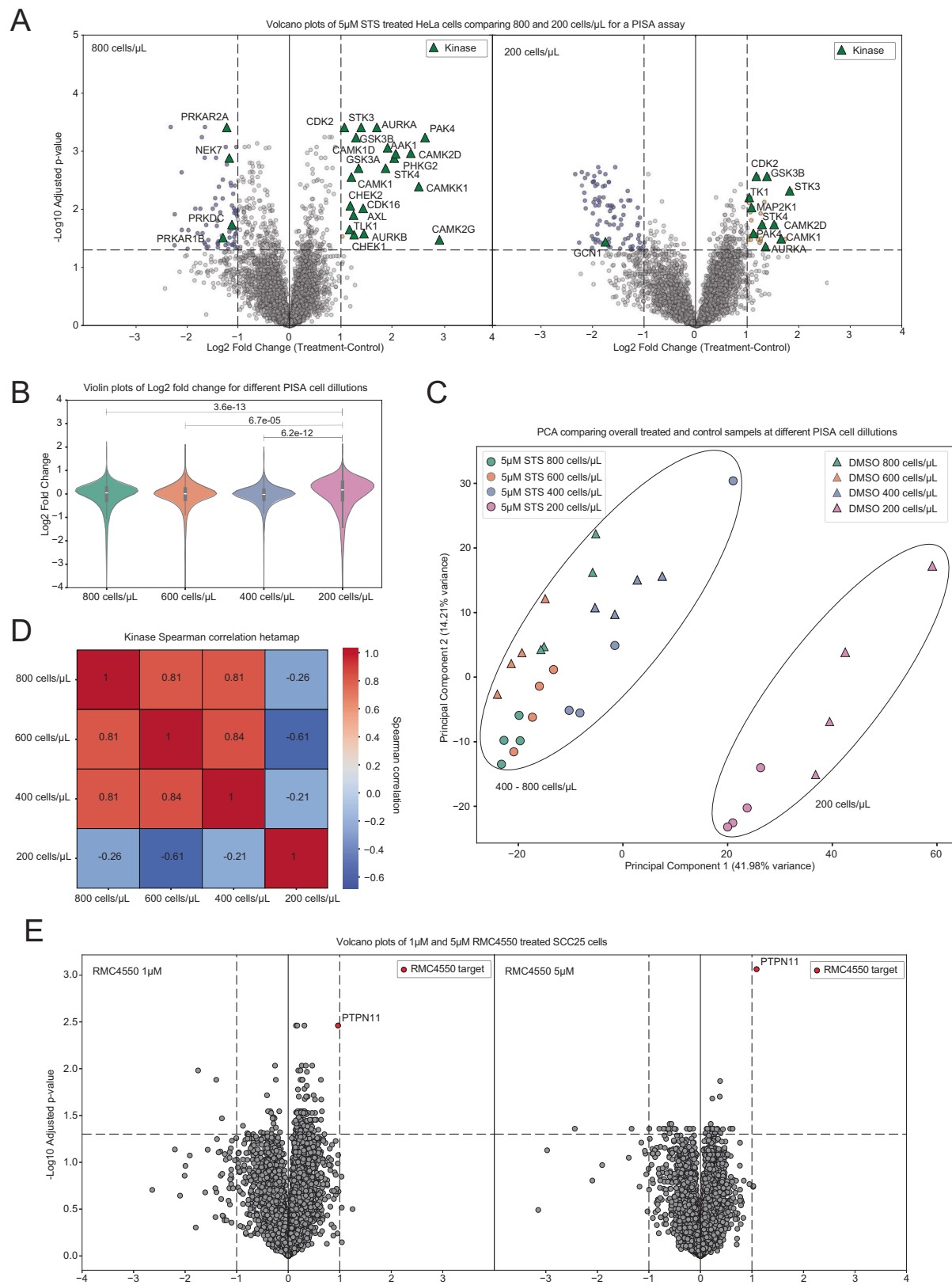

## Comparison of One-Tip-PISA to PISA-PAC

PAC using magnetic microbeads is a well-established method for protein extraction in SDS buffer, followed by efficient detergent removal and on-bead trypsin digestion[21]. As presented by Kverneland et al., it is possible to fully automate the PAC workflow with up to 192 samples processed in parallel within 6 h utilizing an Opentrons OT-2 liquid handling robot[22]. By contrast, the One-Tip-PISA method reduces the digestion time to 2–4 h at 37°C and combines it with sample cleanup, which allows preparing multiple 96-tip boxes in parallel. First, to compare these two methods in terms of quality of analysis, HeLa cells were treated with 5 μM and 10 μM STS, as well

**Fig. 3 | Determination of the low sample limit for One-Tip-PISA. A** Volcano plots comparing 5 µM STS treatment at 800 cells/µL vs 200 cells/µL, with log2 FC and −log10 BH-adjusted *p*-values plotted. Significant kinases meeting adjusted *p* < 0.05 and log2 FC > 1 or < -1 are marked as green triangles, with significant proteins shown in blue for negative FC and yellow for positive FC. **B** Violin plot of log2 FC in 5 µM STS treatment across cell dilutions, with boxplots indicating the median, first, and third quartiles within each violin. **C** PCA plot showing explained variance of the first two principal components for median-normalized, log2-transformed intensities, comparing all cell dilutions under 5 µM STS treatment and DMSO control. Ellipses highlight the distinct clustering of samples at 200 cells/µL vs other dilutions. **D** Heatmap showing the Spearman correlation of the fold-change for kinases with significant changes in stability (BH-adjusted *p* < 0.05 and log2 FC > 1 or < −1), across the four cell dilutions against the respective DMSO controls. **E** Volcano plots comparing 1 and 5 µM treatment of SCC25 cells with SHP-2 inhibitor RMC4550, with log2 FC and −log10 BH-adjusted *p*-values plotted. Highlighted is the target PTPN11 (SHP-2) as a red dot.

as DMSO (vehicle), and diluted to cell concentrations ranging from 800 cells/µL to 200 cells/µL. The PISA assay was performed, after which the samples were split into two groups: one processed via PAC and the other via One-Tip-PISA. First, we compared the CV between the methods under different treatment conditions, as shown in Fig. 4A. Across all cell dilutions and treatments, the One-Tip-PISA method consistently yielded smaller CV values for the identified kinases, suggesting higher reproducibility. This is also true for the overall CV values considering all proteins (Supplementary Fig. 7). However, comparing these CV values from Fig. 4A obtained with One-Tip-PISA to CV values from Ye et al. obtained by only utilizing One-Tip without the PISA assay shows an increase in sample variance by up to 20%[18]. This difference can be explained by the several extra steps that a PISA assay incorporates this including more pipetting steps from splitting and pooling sub-samples from 10 temperature points after the heat treatment in the PCR instrument, as well as removal of supernatant after centrifugation.

Next, we investigated protein subcellular localization using DeepLoc 2.0 (Fig. 4B) as described above. The 200 cells/µL samples treated with 5 µM STS, 10 µM STS, or DMSO for both One-Tip and PAC preparation show clear differences compared to the other groups. To evaluate these differences, a one-way ANOVA was performed on each subcellular localization, grouping the samples by cell concentration (Supplementary Table 2). The ANOVA results reveal significant differences for proteins localized in the cytoplasm, mitochondria, and for proteins shared between the cytoplasm and nucleus. Other localizations did not show significant variance either within or between groups. The analysis suggests that while both methods are generally consistent, certain localizations show more variability depending on the method used.

Lastly, we compared the relative abundance of kinase detection across both methods using Spearman correlations of detected kinases after a 5 µM STS treatment with One-Tip-PISA and PAC (Fig. 4C). Overall, the correlation between PAC and One-Tip-PISA remains high, between 0.71 and 0.76. However, as the cell concentration decreases, fewer kinases can be used for correlation. Interestingly, PAC detects significantly more kinases than One-Tip-PISA, particularly at 800 and 600 cells/µL. This difference may be attributed to the distinct sample preparation workflows of the two methods. For One-Tip-PISA, 5 µL of lysed cell supernatant is directly loaded onto the Evotip, where digestion is performed above the stationary phase of the tip itself. Since this amount of protein and sample volume is too low for PAC digestion, we had to load the entirety of the samples. Thereafter, the samples were digested on beads prior to peptide loading, with approximately 25 ng of peptides subsequently loaded onto each Evotip. Therefore, while the peptide amount injected into the LC-MS was roughly the same, the amount of protein used for digestion was vastly different. This is reflected in the digestion efficiency, which is likely to differ between the on-tip digestion in One-Tip-PISA and the overnight digestion with the PAC method. This may also contribute to variations in the detection of kinases. Despite these differences, One-Tip-PISA demonstrates a strong correlation with PAC and provides a more streamlined workflow with higher throughput, while PAC excels in detecting a greater number of kinases in larger cell number samples.

This is not only true for kinases but also for all proteins in general (Supplementary Fig. 8), with correlations between all cell dilutions comprised between 0.8 and 0.84, and considering 3200–3900 proteins. Although the PAC method demonstrates an advantage in uniquely identifying kinases within the 800–600 cells/µL range compared to the One-Tip-PISA method, it is crucial to note that not all sample populations can be extended to such input levels. For instance, cancer stem cells, which are extremely relevant for anti-cancer treatments and usually represent a small subpopulation within a larger cancer cell population, must often be FACS-sorted, resulting in low cell concentrations (e.g., $5 \times 10^6$ cells/mL)[23]. Such samples are typically subdivided into multiple replicates and treatment conditions, emphasizing the need for methods that can accommodate very low cell inputs. Similarly, the analysis of freshly isolated murine immune cells, where approximately $3 \times 10^5$ cells can be retrieved from 2 µg of mouse tissue via FACS sorting, also highlights the importance of approaches tailored for minimal sample input[24]. Moreover, expensive and labor-intensive cells, such as induced pluripotent stem cells (iPSCs), serve as important models for kinase inhibitor studies but require significant expertise and resources for cell cultivation[25]. As proteomics continues to advance and mature towards clinical and biomedical applications, the ability to analyze low-input samples becomes increasingly important, to ensure that limited yet biologically valuable material can be effectively utilized with optimized and streamlined methods[26].

## High-throughput, 96-well plate format approach

To showcase the suitability of the One-Tip-PISA workflow for high-throughput studies, we employed a 96-well plate format and tested a panel of 10 kinase inhibitors and one tyrosine phosphatase inhibitor in two concentrations (1 µM and 5 µM), as well as a DMSO control each in four replicates. The entire workflow from drug treatment to injection into the mass spectrometer was performed within a day. HeLa cells were treated with Erlotinib, Sorafenib, RMC4550, BIRB760, BIO (6-bromoindirubin-3'-oxime), SP600125, MK-2206, Torkinib, Akt1/2, Wortmannin and Enzastaurin. The number of proteins quantified in the analysis was between 6736 and 7091 proteins on average (Fig. 5A). Next, we plotted the CV of protein abundances for each condition, where all 23 display a median CV value below 0.25. (Fig. 5B). The inhibitors are all targeting kinases of the EGFR signaling pathway, except RMC4550, which targets the tyrosine phosphatase SHP2 (Fig. 5C).

To visualize the different target profiles of each inhibitor, we generated a heatmap representing the log2 fold-change of all significant kinases detected in all treatments compared to the control (Fig. 5D). All treatments significantly impacted the thermal stability of kinases, except Erlotinib and Torkinib. Sorafenib is a multi-kinase Raf inhibitor, and *RAF1* was one of the main targets of this inhibitor, highlighted in the heatmap, as well as the known target *MAPK14*[27,28]. Erlotinib is a tyrosine kinase inhibitor that targets *EGFR*. However, based on the heatmap, no known target or off-target could be identified among the kinases *AGK*, *AKAP1*, *AURKB*, *CSNK1G3*, and *PGK2*, which show a concentration-dependent stability alteration. EGFR, the main target of Erlotinib, has been detected in the dataset (Supplementary Fig. 9); however, EGFR was not significantly stabilized or destabilized during the PISA assay, neither in 1 µM nor 5 µM treatment concentration. The p38α inhibitor BIRB760 engaged its known target *p38* (*MAPK14*)[29]. Other kinases highlighted in the heatmap, like *AGK*, *AKAP1, AURKB, MAPK8, PAK4*, or *PGK2*, do not have any supporting literature to mark them as potential off-targets for BIRB760. The *GSK3* inhibitor BIO targeted both of its known targets, *GSK3A* and *GSK3B*[30]. For the other kinases with a significant FC and concentration-dependent trend, like AAK1, CDK13, *PAK4, PGK2*, and *STK4*, no literature connection to BIO was found. For the JNK inhibitor SP600125, its known target *MAPK8* was identified in the data; however, only for the 1 µM treatment and not for

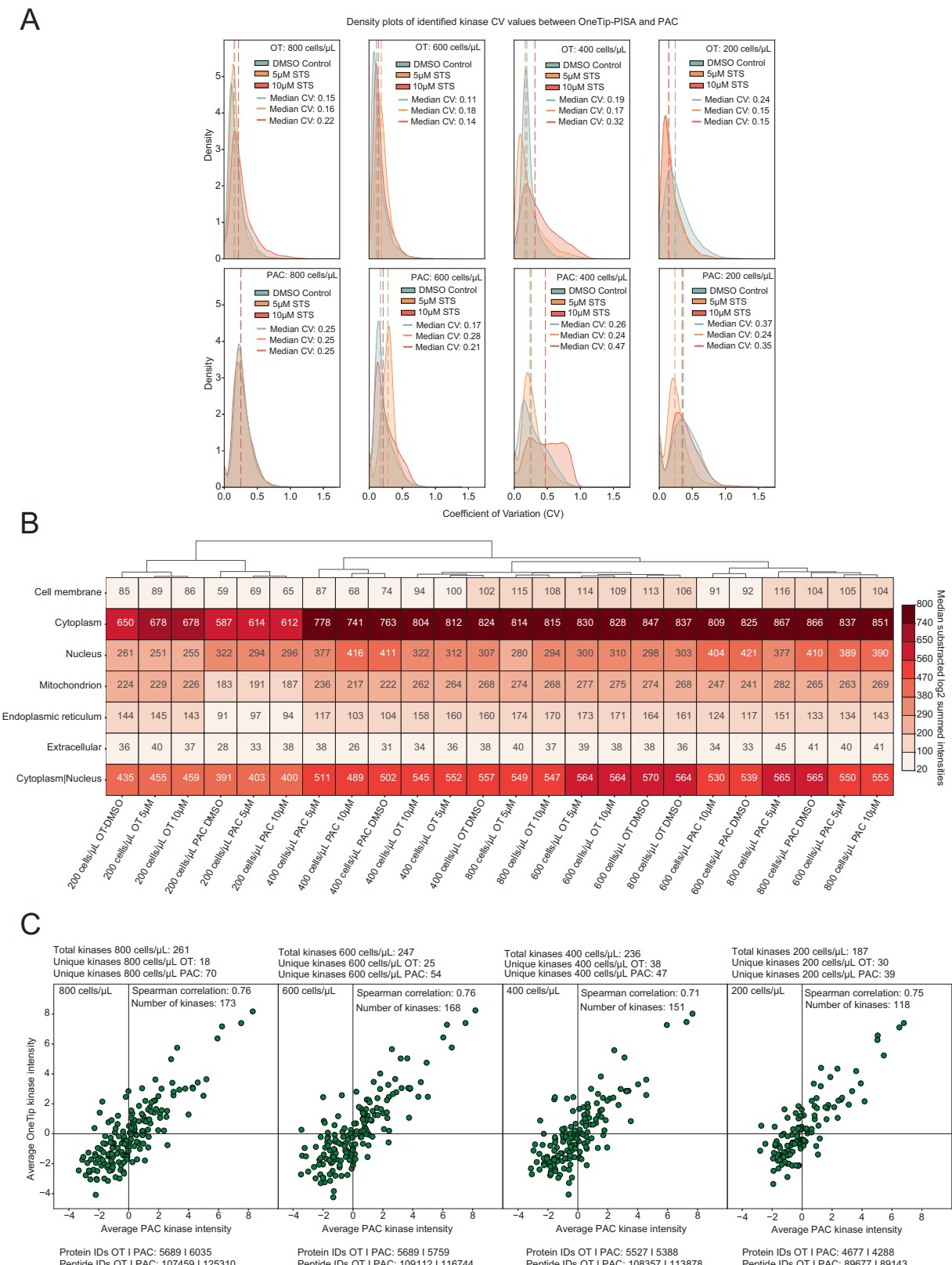

**Fig. 4 | Performance comparison between One-Tip-PISA and PAC.**
**A** Distribution of the CV of all kinases identified using PAC digestion and the One-Tip-PISA approach across cell dilutions from 200 to 800 cells/μL post-PISA treatment; dotted lines indicate the median CV for each distribution. **B** Heatmap with hierarchical clustering of treatments (10 μM and 5 μM STS, DMSO control) across both One-Tip-PISA and PAC workflows at all cell dilutions, showing median-subtracted, log2-transformed summed intensities across cell compartments.

**C** Spearman correlation plots of median-normalized, log2-transformed kinase intensities following 5 μM STS treatment, comparing PAC and One-Tip-PISA workflows across four cell dilutions. Each plot displays the cell number, Spearman correlation coefficient, number of correlated kinases, and unique kinases exclusive to either PAC or One-Tip-PISA, which could not be correlated. Furthermore, the total number of proteins and peptides for One-Tip-PISA and PAC, has been added.

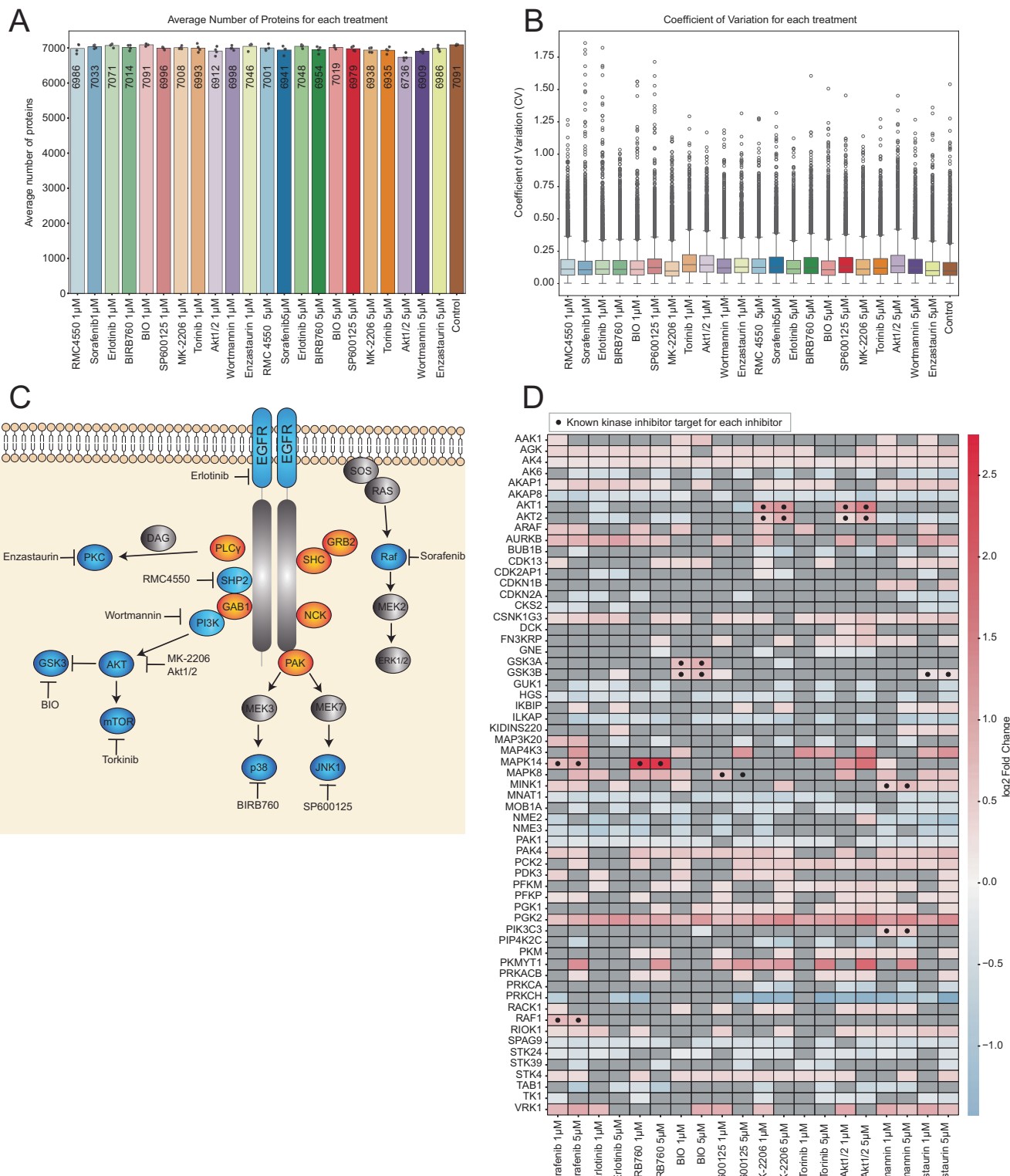

**Fig. 5 | Evaluation of One-Tip-PISA on a 96-well plate approach. A** Bar plot showing the average number of identified protein groups for each of the 11 inhibitors at two concentrations (1 μM, 5 μM), along with the DMSO control; individual replicates are represented as dots ($n = 4$). The treatment of BIO at 5 μM contained one outlier identified via Grubbs-test max $G$-value 1.495 > critical $G$-value 1.481, which was thus removed ($n = 3$). **B** Box plots of the CV across all inhibitors, concentrations, and DMSO control. Each box plot shows the median (black line), with first and third quartiles and whiskers representing the full data range. **C** Visual representation of the EGFR signaling pathway and its involved kinases, as well as the used inhibitors and their target. **D** Heatmap of log2 FC for kinases with significant variance across treatments (based on one-way ANOVA) and concentration dependence between 1 μM and 5 μM treatments; black dot indicates inhibitors that interact with known kinase targets, only kinases plotted with a minimum |FC| > 0.25.

the 5 µM treatment[31]. Other kinases like *AGK, CSNK1G3, PAK4, PGK1, PGK2, PKMYT1*, and *STK4* showed a significant FC, as well as a concentration-dependent effect between 1 and 5 µM treatments, while having no known connection to the inhibitor. The well-characterized AKT inhibitor MK-2206 showed high specificity in its target engagement, where *AKT1, AKT2* show significant shifts in their FC in 1 µM and 5 µM treatments[32]. Further kinases that show significant shifts in FC are *AGK, AKAP1, PAK4, PCK2, PDK3, PGK1, PGK2, PKMYT1, RACK1*, and *STK4*, but these kinases have no known connection to the AKT inhibitor. The m-TOR inhibitor Torkinib showed a significant shift in *AKAP1, CSNK1G3, MAP4K3, PCK2, PGK2*, and *STK4* kinases. While mTOR, the actual target of Torkinib, was detected in the dataset, it was not identified as statistically significant based on the FC between treatment and control. Similarly, its downstream molecule *EIF4BP1* was also detected but did not show significance in FC or *p*-value. As shown in Supplementary Fig. 10[33] Akt1/2 targets the kinases *AKT1* and *AKT2*, and both targets showed a significant increase in stability[34]. Other significant kinases include *AK4, CSNK1G3, DCK, FN3KRP, MAPK14, PCK2, PFKM, PFKP, PGK1, PGK2, PKM, PRKACB, RACK1* and *RIOK1* which are not known to be targeted by AKT1/2. Wortmannin is a potent inhibitor of phosphoinositide 3-kinases (PI3Ks). and *PIK3C3* shows significant stability alteration. While not a known target of Wortmannin, it was previously highlighted in the literature as being involved in the drug mechanism of action[35]. Other PI3K family kinases were removed from the dataset due to missing values in some replicates at the highest Wortmannin concentration. However, *PIK3CB* and *PIK3CA* show a strong concentration-dependent destabilization trend upon Wortmannin treatment (Supplementary Fig. 11), resulting in a decreased measurement of protein intensity at 5 µM, which could be responsible for the missing values at this concentration. This suggests that rather than stabilizing the protein complex, Wortmannin may have destabilized it, reducing its solubility compared to the DMSO control. A Co-crystallization study showed that Wortmannin binding induces conformational shifts in kinase domain loops in *PIK3CB* and *PIK3CA*, affecting the ATP-binding pocket[36]. The inhibitor Enzastaurin is designed to inhibit *PKCβ*, but this kinase is absent from the dataset, while *GSK3B* was identified as significant and is a known off-target of Enzastaurin. This suggests that the treatment was effective; however, PISA may not be able to solubilize the inhibitor-target complex with *PKCβ*, which could explain its absence in the dataset[37]. Although RMC4550 is not included in the heatmap since SHP-2 is not a kinase, it was found to bind its known target, *PTPN11* (SHP-2), as shown in Supplementary Fig. 13A. The log2 ratios of treatment vs control indicate a concentration-dependent effect between the DMSO control, 1 µM and 5 µM which have been underlined by a Welch's *t*-test with *p*-values of 0.0528 (1 µM) and 0.0169 (5 µM).

## High throughput One-Tip-PISA approach on SCC25 cells with kinase and non-kinase inhibitors

To expand the scope of the One-Tip-PISA workflow, we also applied it to a second cell line, the epithelial-like SCC25 cells. These cells were treated with two different kinase inhibitors, Erlotinib and Enzastaurin, as well as two non-kinase inhibitors, RMC4550 and Thapsigargin, at 1 µM and 5 µM treatment concentrations with 4 replicates. In the same way as for the HeLa treatments, a heatmap with significantly changed proteins based on a one-way ANOVA that also shows a concentration-dependent effect was plotted. The heatmap in Fig. 6A shows that the allosteric SHP2 inhibitor RMC4550 significantly stabilized its known target *PTPN11* (Supplementary Fig. 13A). For the tyrosine kinase receptor inhibitor Erlotinib, *EGFR* was not highlighted as statistically significant, thus showing the same result as for the HeLa Erlotinib treatment. It is known that SCC cells overexpress *EGFR*, and by utilizing SCC cells, we hoped to see a stronger interaction between the inhibitor and its tyrosine kinase target[38]. However, in Supplementary Fig. 12, we can see that *EGFR* is present in the dataset but does not cross the significance criteria either in the 1 µM or 5 µM treatment condition. Thapsigargin is an endoplasmic reticulum $Ca^{2+}$-ATPase inhibitor targeting *SERCA*. *SERCA2A* was not identified in the dataset. Whilst with new computational approaches to TPP like InflectSSP, it was shown that

*SERCA2A* can be determined as a target, this was not the case for our dataset[39]. We found two known downstream targets of Thapsigargin treatment, which showed significantly altered stability (highlighted with black dots on the heatmap in Fig. 6A) (Supplementary Fig. 13B, C). The first one is the c-fos protein, which is known to be affected by Thapsigargin treatment as part of its early cellular response, likely mediated by calcium signaling disruptions[40]. The second protein is Transthyretin (*TTR*), a tetrameric transport protein that is affected by induced ER stress and impairs the folding of *TTR*. This could explain its decrease in stability compared to the DMSO control[41]. Lastly, Enzastaurin, the *PKCβ* inhibitor, shows no significant stabilization of *PKCβ*, similarly to the treatment of HeLa cells. No published study to date has reported a strong detectable thermal stabilization of *PKCα/β* by Enzastaurin. However, this does not necessarily mean that Enzastaurin fails to bind *PKCβ* in cells. It is likely that the interaction may not produce a large thermal shift under our assay conditions, a known limitation of the TPP/PISA approaches. Both Dart et al. and Owens et al. describe the presence of false negatives in their thermal shift assay-based methods and thus highlighting the importance of orthogonal methods like phospho-proteomics or affinity pull-downs to evaluate drug target binding[42][,43]. This can originate from the structure of *PKCβ*. Indeed, *PKCβ* (~77 kDa) has a large, multi-domain structure, and binding to its kinase domain may not sufficiently stabilize the full kinase to produce a thermal stability shift. Thus, the lack of a significant thermal stability shift can be attributed to inherent assay limitations and not to the absence of binding. Using complementary methods like affinity pull-downs can help solidify evidence of the binding of a small inhibitor drug to its kinase target.

Finally, while *PKCβ* was not stabilized by Enzastaurin, we detected a significant stability shift for GSK3B. Enzastaurin was not intended to target *GSK3B*, but Reinecke et al. show by utilizing kinobeads that the inhibitor has a CATDS (concentration and target dependent selectivity) value of 0.25 towards GSK3B. This demonstrates that Enzastaurin exhibits a moderate selectivity against *GSK3B*[44] and confirms our results. Figure 6B illustrates a subset of the data, including only proteins with an FC above 0.5, indicating increased stability compared to the DMSO control. The stacked bar plot categorizes the identified proteins based on their predicted subcellular localization, as determined by DeepLoc 2.0. The localization categories include the cell membrane, cytoplasm, endoplasmic reticulum, extracellular space, mitochondrion, and nucleus. Additionally, proteins containing predicted signal peptides are highlighted. Across the four inhibitor treatments, the total number of stabilized proteins varies, with Enzastaurin yielding the highest count (153 proteins) and Erlotinib the lowest (70 proteins). Interestingly, despite this variation, the relative proportion of membrane proteins remains relatively consistent across treatments, ranging between 5% and 9% of the total identified proteins, however, no membrane-bound kinase has been identified in the membrane protein compartment. The same is true for the other subcellular localizations, as the overall percentage values of the localization do not vary between treatments. Especially overrepresented are the extracellular proteins and identified signal peptides from the Thapsigargin treatment. Interestingly, the protein *TRAM2* has been identified in the ER fraction, a protein known to interact with the ER $Ca^{2+}$ pump *SERCA2B*, which is impaired by Thapsigargin[45]. Figure 6C displays a shift in Cytoplasm proteins showing higher proportions in Enzastaurin (32%) compared to the stabilized proteins of the Enzastaurin treatment (19%). For Erlotinib, three times the number of membrane proteins have been identified compared to the stabilized region, however, no membrane kinases have been identified within these 16 proteins. For the RMC4550 treatment, we see an increased number of ER proteins in the destabilized region compared to the stabilized region. Also, the number of extracellular proteins destabilized is 14, more than three times as large as for the stabilized subset, with only 4. Lastly, Thapsigargin displays only half as many proteins with an FC below -0.5 compared to the stabilized counterpart; however, the number of signaling peptides is still the highest within the group of 4 inhibitors, the same as having the highest number of extracellular proteins in addition to RMC4550. These findings highlight that One-Tip-PISA effectively recovers proteins from a broad range of subcellular localizations without bias toward

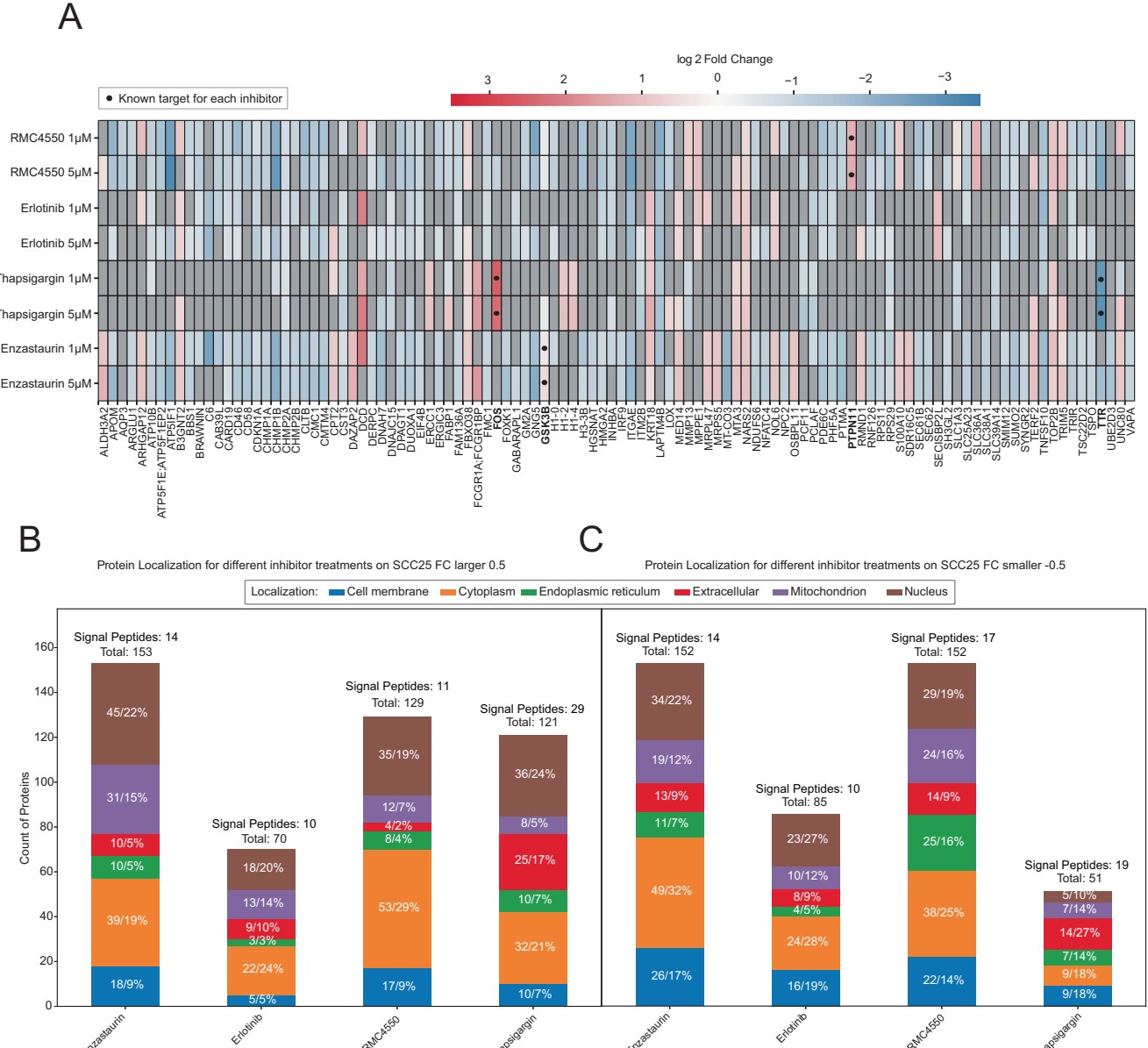

**Fig. 6 | Evaluation of One-Tip-PISA SCC25 Cells. A** Heatmap of log2 FC for proteins with significant variance across treatments (based on one-way ANOVA) and concentration dependence between 1 μM and 5 μM treatments; black dot indicates known inhibitor targets. The only proteins plotted had a minimum |FC| > 0.5 between treatment and control. FC were calculated as log2-transformed and median-normalized protein intensities under treatment conditions and their DMSO control. **B** Stacked bar plots of proteins with FC > 0.5, representing proteins stabilized during the PISA assay following treatment with Erlotinib, Enzastaurin, RMC4550, and Thapsigargin in both treatment concentrations. Proteins are categorized by subcellular localization (DeepLoc 2.0), with percentage values contextualized to all proteins in this subset. Signal peptides and the total number of proteins are highlighted above each bar. **C** Stacked bar plots of proteins with FC < −0.5, representing destabilized proteins across treatments. Protein subcellular localization is highlighted, with relative proportions provided for each category.

any specific compartment. The distinct shifts in protein distribution between stabilized and destabilized subsets across different inhibitors further emphasize the method's sensitivity to diverse cellular responses even under low sample input, which allows for in-depth analysis of subcellular compartments.

## Conclusion

We have shown that the One-Tip-PISA method is a reproducible, streamlined, and sensitive thermal proteomics method allowing for easy processing of intact cells in a 96-well plate format from the PISA assay to processable data within 2 days. We have shown that PISA in combination with One-Tip digestion allows for the processing of limited sample amounts as low as 400 cells/μL, equivalent to 50,000 cells, and in some cases even as

low as 200 cells/μL. This could be further developed and miniaturized as the volume in the tube for each temperature treatment was 12.5 μL, leaving ample room for reducing the volume and thus the number of cells needed for analysis. Comparing the One-Tip-PISA method to the established PAC-based PISA workflow showed a good agreement between the two methods for detecting kinase targets across several cell concentrations. However, One-Tip-PISA displayed smaller CV values between replicates, indicating that increased reproducibility is achievable with a simplified workflow that is less reliant on pipetting. Further advantages are the higher throughput of the One-Tip-PISA, as well as the absence of specific sample preparation equipment, which makes this method substantially more affordable and attractive for laboratories on a smaller budget without access to expensive automation equipment such as a liquid handling robot. Lastly, using a

**Table 1 | Cell stock solution and dilutions used in the experiments**

| Stock solutions [cells/ml] | Cells in 150 µL PBS | Cells per PISA 12.5 µL PCR tube | Cells/µL |
|---|---|---|---|
| 800,000 | 120,000 | 10,000 | 800 |
| 600,000 | 90,000 | 7500 | 600 |
| 400,000 | 60,000 | 5000 | 400 |
| 200,000 | 30,000 | 2500 | 200 |

kinase and non-kinase inhibitor panel, we demonstrated the scalability of the methods by processing 24 different sample conditions in 4 replicates in a 96-well plate format from PISA assay to injection with high analytical depth, as well as low sample variability, emphasizing the speed and ease of use of One-Tip-PISA.

## Materials and methods
### PISA protocol
HeLa human cervix carcinoma cells and SCC-25 human squamous cell carcinoma were cultured in DMEM (Gibco), with the addition of 10% fetal bovine serum, 1% Penicillin-Streptomycin at 37 °C, in an incubator with 5% $CO_2$ atmosphere until 80% confluence and subsequently treated with either an inhibitor (STS, Erlotinib, Enzastaurin, Wortmannin, MK-2206, AKT1/2, BIO (6-bromoindirubin-3'-oxime), Torkinib, BIRB760, SP600125, Sorafenib, RMC4550 and Thapsigargin from Selleckchem) or an equivalent volume of vehicle (DMSO) as control. After the treatment time, the cells were washed with Phosphate Buffered Saline (PBS) (Gibco) and detached using TrypLE (Gibco). The cell detachment reaction was stopped by using fresh DMEM medium; afterwards, the cells were pelleted at $400 \times g$ for 3 min. This step was repeated twice more to rinse the cells, after which the cells were counted with a Cytosmart Exact FL. Based on this count, all HeLa cell dilutions were prepared with 800, 600, 400, and 200 cells/µL for subsequent PISA analysis as described in Table 1. The 12.5 µL of each cell suspension was then equally distributed between ten PCR tubes/96-well PCR plates with four technical replicates for each condition. Cells were heated in a temperature range from 48 °C to 59 °C and later combined again. DDM was subsequently added to all combined samples to a total amount of 0.2% per sample, and the samples were lysed via freeze-thawing in five cycles. Finally, all lysates were transferred and centrifuged at $13,800 \times g$ for 30 min at 4 °C ($4000 \times g$ for the 96-well approach). The supernatant was removed from each sample and collected separately, after which a (BCA) protein assay kit (ThermoFisher Scientific) was used to evaluate the protein content of each sample. Subsequent 5 µL of the supernatant of each sample has been used in the one-tip workflow for digestion.

### One-Tip protocol
Evotips were used for this workflow and prepared according to the vendor's instructions. Dry Evotips were washed with 20 µL of Solvent B (0.1% FA in acetonitrile) and centrifuged at $800 \times g$ for 60 s. Then, the Evotips were soaked in 100 µL of 1-propanol for conditioning until they turned pale white. Lastly, the Evotips were equilibrated with 20 µL of Solvent A (0.1% FA in water) and centrifuged at $800 \times g$ for 60 s. Next, 5 µL of lysis and digestion buffer was pipetted into the Evotips. Where applicable, the PISA supernatant has been diluted to a protein concentration of 5 ng/µL to not oversaturate the column. The buffer contained 0.2% DDM, 100 mM TEAB, 20 ng/µL Trypsin, and 10 ng/µL Lys-C. After the PISA supernatant and digestion buffer were pipetted into the Evotips, the tips were briefly centrifuged at $50 \times g$ to mix the digestion buffer and sample. These Evotips were then digested for 3 h in an Evotip box filled with water to the level of the C18 resin and incubated at 37 °C. The reaction was quenched by adding 50 µL of Solvent A to the Evotips and centrifuging for 60 s at $800 \times g$. Thereafter,

samples were washed with 20 µL of Solvent A and centrifuged for 60 s at $800 \times g$. Lastly, 100 µL of Solvent A was added to the tips and centrifuged for 10 s at $800 \times g$ to keep the tips wet.

### PAC protocol
PAC was applied based on the following protocol[21]. HeLa cells were grown and treated in the same way as described before in the PISA protocol. Based on the protein measurement after the cell lysis and removal of supernatant with the ThermoFisher BCA kit, MagReSyn® Hydroxyl beads (Resyn Biosciences) were added (2:1, beads/protein, w/w), followed by 700 µL of 100% acetonitrile. PAC digestion was conducted on a KingFisher Flex. The beads were rinsed 3 times with 100% acetonitrile, and 2 times with 70% ethanol, and on-bead digestion was performed in 50 mM ammonium bicarbonate containing LysC endopeptidase (1:500, enzyme/protein, w/w) and trypsin (1:250, enzyme/protein, w/w). Samples were acidified using formic acid to a pH < 3 and 25 ng of peptide was loaded onto Evotips prior to LC-MS/MS analysis.

### LC-MS/MS
LC-MS/MS analysis was performed on an Orbitrap Astral (ThermoFisher Scientific) mass spectrometer using Tune software (version 1.0) coupled to an Evosep One system (EvoSep Biosystems). Samples were analyzed using the Zoom Whisper mode in 40SPD (31-min gradient) and 80 SPD (16-min gradient) using a commercial analytical column (15 cm Aurora Elite TS, IonOptics) or a 5-cm Aurora Rapid column, respectively. The column was interfaced to the MS using an EASY-Spray™ source. The Orbitrap Astral MS was operated at a full MS resolution of 240,000 with a full scan range of 380–980 m/z when stated. The full MS AGC was set to 500%. The nDIA-MS/MS scans covering m/z range of 380–980 were recorded with 4Th isolation windows and 6 ms maximum ion injection time (IIT) for all samples. The isolated ions were fragmented using HCD with 25% NCE. In the case of SCC25 treatment of RMC4550 and Thapsigargin, a 60 SPD (21 min gradient) was used.

### MS data analysis
Raw MS files were analyzed in Spectronaut v19 (Biognosys) with a spectral library-free approach (directDIA+) using the protein reference database (Uniprot 2022 release, 20,598 sequences) complemented with common contaminants (246 sequences). This method does not utilize reduction and alkylation, database searches were performed with free cysteine sulfhydryls, and hence cysteine carbamylation was not set as a fixed modification, whereas methionine oxidation and protein N-termini acetylation were set as variable modifications. Cross-run normalization was not checked. Each experiment with different conditions of DDM concentrations, cell numbers, as well as Inhibitor/Control treatment was analyzed separately by searching them with the method evaluation mode enabled. Each sample was run with four replicates, and only the 0.6% DDM condition lacked one replicate that was lost during sample preparation. The extracted data was analyzed and visualized with Python 3.11.0 in Visual Studio Code, utilizing GitHub Copilot for code-related questions.

### Reporting summary
Further information on research design is available in the Nature Portfolio Reporting Summary linked to this article.

## Data availability
The data can be accessed on PRIDE via the following accession numbers: Project accession: PXD056894 and PXD061105.

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

## Acknowledgements

Work at the Novo Nordisk Foundation Center for Protein Research (CPR) is funded in part by a donation from the Novo Nordisk Foundation (NNF14CC0001, NNF24SA0098829, and NNF21OC0072070). J.V.O. is also funded by Novo Nordisk a/s (CELFFI-2022-002843). P.S. is funded by the Swedish Research Council (2022-00323).

## Author contributions

P.S. and J.V.O. conceptualized the study. M.L. performed the experiments. M.L., P.S., and J.V.O. analyzed the data. M.L., P.S., and J.V.O. wrote the article. All authors read, edited, and approved the final version of the manuscript.

## Competing interests

The authors declare no competing interests.
