## [Peer review file · Communications Chemistry]

High-throughput proteome integral solubility alteration assay for low cell input using One-Tip

Corresponding Author: Professor Jesper Olsen

Version 0:

Reviewer comments:

Reviewer #1

(Remarks to the Author)

In this manuscript, the authors aim to develop the One-Tip-PISA method by combining PISA with the One-Tip technology to minimize sample handling and reduce the need for large input amounts for drug-target profiles. The authors first indicated the advantage of DDM over PBS in kinase detection, and evaluated the effects of varying DDM concentrations on cell lysis, ultimately selecting a 0.2% DDM concentration for their studies. The One-Tip-PISA method exhibited more potential than the traditional PISA-PAC approach especially for low cell input, of which sensitivity limit reached 200 cells/ μ L. Furthermore, the authors adapted the One-Tip-PISA method to a 96-well plate format, utilizing a kinase inhibitor panel to profile both known targets and off-targets successfully. Overall, this novel integration of One-Tip-PISA is not only reproducible and efficient but also highly sensitive, allowing for an easy processing of intact cells in a 96 well format. Specific comments are listed as following:

Major comments:

1. According to Reference 11, One-Tip technology has already been applied for single-cell proteomics, and showed high reproducibility and quantitative precision, with a remarkable coefficient of variance of 5–8% (3000 cells to 1000 cells). However, in this work, as shown in Figure 5A, the CV value was 30% for 2500 HeLa cells (200 cells/ μ L) treated by DMSO (vehicle). What accounts for this kind of variance?
2. In line 262, PAC detects significantly more kinases than One-Tip-PISA, particularly at 800 and 600 cells/ μ L. It seemed that PAC was better for kinases identification if low cell input (200 cells/ μ L) was not necessary. Therefore, it is essential to provide more discussion to emphasize the significance of low cell input in PISA, which will make it clear why such a threshold is necessary and how it enhances the applicability of One-Tip-PISA for limited cells.
3. In Figure 5, when comparing these two methods (One-Tip-PISA and PISA-PAC), the number of identified proteins and peptides should be given for each treatment.

Minor points:

1. In line 87, I didn't find any discussion about 1%DDM from Reference 14, please check it.
2. In Figure 3A, the deviation of 400 cells/ μ L (10 μ M STS) was too large that the BCA assay might not be suitable for such a low concentration of cells treated by STS.
3. In line 263-265, 5 μ L of lysed cell was loaded for One-Tip-PISA while 25 ng of peptide was loaded for PAC-PISA. The sample loading was different? If the initial processing amount was equivalent, please make it clear.
4. Please correct the number in figure legends of Figure 3~5.

Reviewer #2

(Remarks to the Author)

This work establishes a method for streamlined sample preparation for PISA samples with post-heat treatment processing using a One-Tip method on an Evosep Evotip cartridge. The research team also evaluated input minimums for the PISA method for acquisition with DIA on an Orbitrap Astral. A limitation of this study is the focus on staurosporine-treated HeLa cells. From a benchmarking perspective, staurosporine has been used for multiple previous CETSA/PISA studies. The benefit to that is the ability to compare these datasets to many other datasets for comparison of proteome depth of coverage, number of positive target ids, and utility of the DIA method. The limitation is that staurosporine is an extremely broad, low specificity, kinase inhibitor. As such, there are numerous positive hits giving a possibly overly optimistic impression of assay sensitivity for this new method. If more selective inhibitors are used, the ability to use low input material for the PISA studies has not been established with this work. The positive aspects of this work are the integrated samples preparation method in a 96-well format, although that is not an entirely unique approach since it builds on a large foundation of previous work. The data from the 96-well inhibitor screen are interesting but are underdeveloped relative to other described findings that have lower overall novelty. The manuscript would benefit from editing to refocus on the 96-well screen with some of the low cell

input datasets being removed or deemphasized.

Specific major comments:

1. It is stated that DDM enhances membrane protein recovery after PISA shown in Fig. 1B and C but should be referring to Figure 2B and C. States a t-test was performed which is not the proper statistical test for protein class enrichment however the figure legend states that it was a 1-way ANOVA. This was a major point in the manuscript and should be clarified and clearly addressed to show the utility of DDM for PISA. Comparisons to membrane protein yields in DOI: 10.1038/nmeth.3652 should be performed. Comparisons to PBS rather than another non-ionic detergent are not informative. Suggest comparison to 0.4% NP40 based on the published work above, although the comparison to PBS can still be included and could be informative to some research groups. It is expected that additional processing steps for samples extracted with 0.4% NP40 would decrease yields but this is not shown. Fig. 2D is not convincing to this reviewer that the effect is specific to 0.2% DDM, could also be a replicate effect. The shape of the volcano plot overall between PBS and DDM is very different, even though the scale is identical. What is the reason for that difference? Fig. 2E data and could be moved from Supp. Figure 2 (although log2 ratio should be defined, EGFR looks destabilized by STS based on the figure title but in text it is stated it is stabilized. Correlation values in Supp Fig 4 are quite convincing of DDS extraction reproducibility. Comparison to 0.4% NP40 would be highly informative. It is no surprise based on prior work that nonionic detergent is beneficial over PBS.

2. Membrane protein SERCA2 as a target ID was shown in DOI: 10.1016/j.mcpro.2023.100630. Profiling of another drug that specifically targets a membrane protein that is not a kinase inhibitor could be informative of the general utility of this approach.

3. Figure 3A(1?) is not informative. If authors would like to include it should be in the supplement. Overall, low input amount experiments with STS are not very novel (Fig. 3 & 4). Lots of PISA and CETSA experiments have been performed with STS and the low cell number data is likely to be noisier and there is no scientific justification for why such low cell amounts need to be used for benchmarking. This comes across as appealing to a trend rather than something with scientific merit. STS is so non-specific that it is easy to call any kinase a target. Overall data not informative in this reviewer's opinion. Who needs to know a lower input limit for easy to grow HeLa cells? Both Figure 3 & 4 are not of broad interest.

4. Comparison of One Tip PISA to PISA-PAC is technically informative, but again provides few biological insights. Data is clear and highly correlated across conditions. Again, Figure 5(3) is not highly informative but does provide some technical insights and benchmarking giving more novel analytical insights than provided by Figures 3 & 4.

5. Figure 6 has the most interesting data for this manuscript, and it has been heavily summarized. Drugs used are inhibitors of the EGFR pathway. The description of the results is fine, but discussion on why known target kinases are not identified as altered is needed. Additional analysis showing if targets were not found to be stabilized/alterd because they were not detected should be included. Description of the number of membrane proteins and membrane associated kinases should be included. These aspects of the study would be much more informative in the 96-well screen than in the earlier sections focused only on the highly studied STS. Suggest that the analysis and figures of this section be expanded to provide more molecular insights including individual target measurement reproducibility across different treatment conditions. Would be interesting to see kinase performance across the panel of inhibitors and the overlap of significant hits since there was pathway justification for the inhibitor selection.

Minor comments:

1. Axis labels should include the conditions compared in the volcano plots for log2 fold-change.

2. Sentence: "The heatmap shows that across all subcellular fractions, ...endoplasmic reticulum; lysosomal, vacuolar, and mitochondrial showed" should be -- endoplasmic reticulum, lysosome, vacuole, and mitochondria showed"

3. Figure 3 legend is labeled as Figure 1. Figure 4 legend is labeled as Figure 2. Figure 5 is labeled as Figure 3 – this is disruptive to reading the manuscript.

4. Data in Figure 6A are difficult to read because of the large number of colors used. Recommend reducing the color usage. Stars in Fig. 6D are nearly impossible to see by eye. Recommend using dots rather than stars.

Version 1:

Reviewer comments:

Reviewer #1

(Remarks to the Author)

The authors have addressed my comments.

Reviewer #2

(Remarks to the Author)

Overall: This work establishes a method for streamlined sample preparation for PISA samples with post-heat treatment

processing using a One-Tip method on an Evosep Evotip cartridge. This includes assessment of assay performance in a high throughput format, which will be highly useful for the larger drug-target interaction analysis research community. The authors have made many improvements in the manuscript and were very responsive to reviewers' comments. Overall, the manuscript is improved in many ways. Some points remain that should be addressed prior to publication, as discussed below.

Revision points to consider/address:

Figure 2 for the reviewers should be included in the manuscript. It is fine to include this as supplementary material. The advantages of different detergents and/or workflows for membrane protein recovery is of broad general interest. If the figure could be included as a main text figure, that could also be done but that should be up to the authors on where to place these comparisons in their narratives.

Figure 2B (table) needs higher contrast font for improved readability.

Figure 3D: Spearman's correlation is written as speerman correlation. Since the test is a person's name, it should be capitalized in addition to the spelling correction. It is misspelled next to the color scale too.

The addition of the RMC4550 treatments add value to the manuscript with the demonstration of a clear drug-target based stabilization. Volcano plots in Fig. 3E are also much clearer than those presented in Fig. 2C. I would suggest that the plot in 2C be reformatted to match 3E for ease of reading. Text on Figure 2C is also very small.

The rebuttal states that high CV levels are due to PISA sample handling without showing any evidence of that. It is still not clear that the high CVs are due to PISA steps rather than the One-Tip approach. This was not convincingly addressed. It is stated that the CV values differ for various drug treatment conditions in the large-scale screen and that it is "not ideal". Why would there be different CV values across treatment conditions? Treatment conditions should not alter sample preparation reproducibility. This challenge will reduce the number of statistically significant hits.

Please evaluate the manuscript again for typos. See particularly concerning examples such as "howeverwith TPP ewith TPP vwith TPP ewith TPP wwith TPP owith TPP this was not the case for our dataset "

Statement, "Lastly Enzastaurin the PKC β inhibitor shows no significant stabilization of PKC β giving the same results as for treatment on HeLa cells thus implying that this specific drug target complex might not be detectable utilizing a PISA assay." Can this point be expanded? Was this typically shown by TPP or CETSA? Is it fair to say that this drug-target complex is not detectable or that the drug-target complex might not result in a significant change in target protein stability? It is unclear that this is specific to PISA and not the drug-target association. Earlier in the manuscript it is stated that, "PISA may not be able to solubilize the 375 inhibitor-target complex with PKC β , which could explain its absence in the dataset". These two points are inconsistent with each other and need more clarity in both parts of the manuscript.

Supplementary Table 1 and 2 – font is too small to be clearly readable.

Version 2:

Reviewer comments:

Reviewer #2

(Remarks to the Author)

I appreciate all the efforts that the authors put forth to address comments by the reviewers. All of my concerns have been suitably addressed.

Point-by-point rebuttal letter to REVIEWER COMMENTS

Manuscript: **COMMSCHEM-24-0630** by Lechner et al.

Our responses to the reviewers' comments are provided below in **green text**. Key modifications and updates are indicated in the revised manuscript file using track-changes.

General response: We would like to express our gratitude to the reviewers for their valuable feedback on our manuscript. Their constructive criticism and insightful comments have helped us to enhance the quality of our research work.

Below, we have addressed the individual comments raised by the reviewers, point-by-point.

Reviewer #1 (Remarks to the Author):

In this manuscript, the authors aim to develop the One-Tip-PISA method by combining PISA with the One-Tip technology to minimize sample handling and reduce the need for large input amounts for drug-target profiles. The authors first indicated the advantage of DDM over PBS in kinase detection, and evaluated the effects of varying DDM concentrations on cell lysis, ultimately selecting a 0.2% DDM concentration for their studies. The One-Tip-PISA method exhibited more potential than the traditional PISA-PAC approach especially for low cell input, of which sensitivity limit reached 200 cells/L. Furthermore, the authors adapted the One-Tip-PISA method to a 96-well plate format, utilizing a kinase inhibitor panel to profile both known targets and off-targets successfully. Overall, this novel integration of One-Tip-PISA is not only reproducible and efficient but also highly sensitive, allowing for an easy processing of intact cells in a 96 well format. Specific comments are listed as following:

We thank the reviewer 1 for his kind words

Major comments:

1. According to Reference 11, One-Tip technology has already been applied for single-cell proteomics, and showed high reproducibility and quantitative precision, with a remarkable coefficient of variance of 5–8% (3000 cells to 1000 cells). However, in this work, as shown in Figure 5A, the CV value was 30% for 2500 HeLa cells (200 cells/L) treated by DMSO (vehicle). What accounts for this kind of variance?

An explanation of why a higher variance in One-Tip-PISA is observed compared to only One-Tip sample preparation was added to the text. In short, this is inherent to PISA sample preparation which includes low-efficiency cell lysis and many additional pipetting steps and sample handling which can lead to variation. For, instance, extra steps including, sample pooling, pipetting and centrifugation for supernatant removal thus increasing the overall variance.

2. In line 262, PAC detects significantly more kinases than One-Tip-PISA, particularly at 800 and 600 cells/L. It seemed that PAC was better for kinases identification if low cell input (200 cells/L) was not necessary. Therefore, it is essential to provide more **discussion** to emphasize the significance of low cell input in PISA, which will make it clear why such a threshold is necessary and how it enhances the applicability of One-Tip-PISA for limited cells.

More discussion and literature references have been added to emphasise the importance of discrete cell types which are limited in available cell number stressing the importance of sensitive sample preparation methods like One-Tip-PISA.

3. In Figure 5, when comparing these two methods (One-Tip-PISA and PISA-PAC), the number of identified proteins and peptides should be given for each treatment.

In Figure 4 (Previously Figure 5) the protein and peptide numbers have now been added in Figure 5C to compare PAC and One-Tip-PISA directly.

Minor points:

1. In line 87, I didn't find any discussion about 1%DDM from Reference 14, please check it.

The source has been changed, the relevant information can be found in the Supplementary Text and Figures of the article.

2. In Figure 3A, the deviation of 400 cells/L (10 μ M STS) was too large that the BCA assay might not be suitable for such a low concentration of cells treated by STS.

This is correct. The intention with this figure was to give researchers an idea about what protein identification numbers they can expect from different cell dilutions. This figure was moved to supplementary figures because it is of lower importance compared to other figures.

3. In line 263-265, 5 L of lysed cell was loaded for One-Tip-PISA while 25 ng of peptide was loaded for PAC-PISA. The sample loading was different? If the initial processing amount was equivalent, please make it clear.

An explanation to the different sample loads was added in the text to help clarify the differences in the sample loading of One-Tip-PISA (proteins are digested above the stationary phase of the tip) and PAC (Proteins are digested in a 96-well plate and peptides are loaded on EvoTips) and thus the variation and differences in detected kinases.

4. Please correct the number in figure legends of Figure 3~5.

The numbers of the figures in the PDF have been corrected

Reviewer #2 (Remarks to the Author):

This work establishes a method for streamlined sample preparation for PISA samples with post-heat treatment processing using a One-Tip method on an Evosep Evotip cartridge. The research team also evaluated input minimums for the PISA method for acquisition with DIA on an Orbitrap Astral. A limitation of this study is the focus on staurosporine-treated HeLa cells. From a benchmarking perspective, staurosporine has been used for multiple previous CETSA/PISA studies. The benefit to that is the ability to compare these datasets to many other datasets for comparison of proteome depth of coverage, number of positive target ids, and utility of the DIA method. The limitation is that staurosporine is an extremely broad, low specificity, kinase inhibitor. As such, there are numerous positive hits giving a possibly overly optimistic impression of assay sensitivity for this new method. If more selective inhibitors are used, the ability to use low input material for the PISA studies has not been established with this work. The positive aspects of this work are the integrated samples preparation method in a 96-well format, although that is not an entirely unique approach since it builds on a large foundation of previous work. The data from the 96-well inhibitor screen are interesting but are underdeveloped relative to other described findings that have lower overall novelty. The manuscript would benefit from editing to refocus on the 96-well screen with some of the low cell input datasets being removed or deemphasized.

We thank the reviewer for the feedback and constructive comments, and we agree that the focus of the article should be broadened.

Specific major comments:

1. It is stated that DDM enhances membrane protein recovery after PISA shown in Fig. 1B and C but should be referring to Figure 2B and C.

We thank the reviewer for pointing this out. It is correct that Figure 1A and B only show a difference in the number of identified proteins and peptides by using DDM instead of PBS. Therefore, it was not appropriate to refer to these Figures for the higher recovery of membrane proteins, which was introduced in Figure 2 B and C. Consequently, the text has now been changed to make this point more evident.

States a t-test was performed which is not the proper statistical test for protein class enrichment however the figure legend states that it was a 1-way ANOVA. This was a major point in the manuscript and should be clarified and clearly addressed to show the utility of DDM for PISA.

We are sorry about the confusion. The mentioned t-test was referring to figure 1B and C. Where a Welch's t-test was performed to compare the average identified proteins and peptides between different DDM concentrations and the PBS control. In Figure 2B, a 1-way ANOVA was performed and evaluated via a F-Test to determine if there are statistically significant differences in the different cell compartments while using DDM or PBS. The ANOVA results table was moved to the supplementary information and a more detailed explanation has been added to make it more clear to the reader which tests have been performed.

Comparisons to membrane protein yields in DOI: 10.1038/nmeth.3652 should be performed.

We thank the reviewer for this suggestion, which we have followed. The available tables from DOI: 10.1038/nmeth.3652 have been downloaded and compared to two datasets. The one produced in this paper using DDM and a PAC digest (Lechner et al) as well as with doi.org/10.1038/s41467-024-53240-2 (Batth et al) utilizing NP40 with a PAC digest. The results of the analysis can be found in the following plots.

Figure1: Bar chart of qualitative evaluation of identified membrane proteins recovered by using 0.2% NP40 Bath et.al, 0.2% DDM Lechner et.al and 0.4% NP40 Savitski et.al on HeLa cells and DMSO treatment. (Bath et.al and Lechner et.al use PAC digest since OneTip PISA can't be done with NP40 since it is not MS compatible)

Figure 1 compares qualitative number of membrane proteins in each data set. Indicating the using 0.2% DDM is comparable to 0.2% NP40. (Note Lechner et.al uses Orbitrap Astral MS, Bath et.al Orbitrap Exploris 480 and Savitski et.al Q-Exactive Orbitrap with TMT labelling and pre-run fractionation)

UpSet Plot of Proteins with cell membrane localization

Figure 2: Upset-Plot comparing membrane proteins in common and individual to each of the datasets

Figure 2 highlights that with 0.2% DDM and the PAC digest a substantial amount of membrane proteins can be recovered with the largest number of membrane proteins identified in this dataset with

210 whereas Bath et al finds 21 and Savitski et.al 26 specific membrane proteins. Largest overlap is also between Bath et al and Lechner et al 128 shared membrane proteins between the datasets.

Figure 3: Quantitative comparison of log2 summed intensity of all membrane and non-membrane proteins displayed as stack plot between the 3 datasets.

Evaluating the difference between DDM and NP40 shows that there is a similar ratio between membrane and non-membrane proteins while using PAC Bath et al 96.7% to 3.3% and Lechner et al 95.3% to 4.7%. This however is not true for the dataset from Savitski et.al where the ratio is 99.9% to 0.1%

Comparisons to PBS rather than another non-ionic detergent are not informative. Suggest comparison to 0.4% NP40 based on the published work above, although the comparison to PBS can still be included and could be informative to some research groups.

We agree with this statement, however, the purpose of DDM in One-Tip-PISA is a necessity to streamline the method and increase sample throughput, as it is a MS-friendly detergent that is directly compatible with LC-MS. That is not the case for NP40, which needs to be removed prior to MS analysis, f.ex. through a PAC digestion workflow. A comparison between DDM and NP40 on HeLa cells and a PAC digest is displayed on Figures 1-3 above.

It is expected that additional processing steps for samples extracted with 0.4% NP40 would decrease yields but this is not shown.

We agree with the reviewer's statement, but a fair comparison between detergents is not possible as very low sample input with a PAC digest will not work. The scope of the article is not to introduce DDM as a detergent or compare it against other detergents it is a necessary part of the sample preparation to use the One-Tip-PISA method as a MS compatible detergent is mandatory.

Fig. 2D is not convincing to this reviewer that the effect is specific to 0.2% DDM, could also be a replicate effect.

Table 1: Grubbs Test results testing for outliers in 10µM STS in PBS and DMSO control in PBS

Condition	Values	Mean	Standard Deviation	Grubbs' Statistic	Critical Value	Outlier Present
10µM STS in PBS	[5511, 5571, 5318, 5629]	5507.3	135.1	1.401	1.481	FALSE
DMSO in PBS	[5232, 5468, 5084, 5193]	5244.3	161.8	1.383	1.481	FALSE

In Table 1 we tested both datasets 10µM STS in PBS and DMSO control in PBS for outliers to make sure the change in variance is not connected to a replica effect during the experiment. In both cases the Grubbs-test resulted in accepting H_0 that there are no outliers in the dataset.

Figure 4: CV values for each dataset comparing the effects of different DDM concentrations against a PBS control.

Considering Figure 4 we can see that both DMSO PBS, and 10 μ M STS PBS show a larger median CV value than most of the DDM containing conditions thus indicating that the skewed distribution of the volcano plot is a result of the lack of DDM instead of just a replicate issue.

The shape of the volcano plot overall between PBS and DDM is very different, even though the scale is identical. What is the reason for that difference?

-I would argue that this is due to the effect of DDM on stabilising proteins and making sure that they e.g. do not get stuck so much at the surface of a tube. Hence the lower variance in the DDM fraction.

0.2% DDM vs PBS log₂ intensity as distribution of different categories (Histogram) then we see two distributions with and without DDM to convince that there is a quantitative difference

Figure 5: Histograms displaying the log₂ signal intensity against the frequency comparing 10 μ M STS lysed with 0.2% DDM against PBS (left) and DMSO control lysed with 0.2% DDM against PBS (right)

As seen in the article Figure 1B and C a lack of DDM leads to lower overall protein and peptide IDs. Moreover, in Figure 5 the log₂ intensity between PBS and DDM lysed samples show that there is a strong quantitative difference between the two conditions. Samples that have only been lysed with PBS do not result in high signal intensity at a high frequency as often as while using DDM. This implies a lower certainty of quantification and higher variability which is one of the reasons why the volcano plot is skewed as the quantitation accuracy is lower. Furthermore, DDM is known to improve the recovery of hydrophobic proteins by stabilizing them which would be lost in the PBS fraction, thus leading to a larger overall variance in the PBS datasets compared to the ones where DDM is used.

Fig. 2E data and could be moved from Supp. Figure 2 (although log₂ ratio should be defined, EGFR looks destabilized by STS based on the figure title but in text it is stated it is stabilized).

Supplementary figure 2 has been moved from the supplementary to the main panel 2 and the text has been clarified.

Correlation values in Supp Fig 4 are quite convincing of DDS extraction reproducibility. Comparison to 0.4% NP40 would be highly informative. It is no surprise based on prior work that nonionic detergent is beneficial over PBS.

A comparison between DDM and NP40 has been implemented earlier during the revision in Figure 1-3.

2. Membrane protein SERCA2 as a target ID was shown in DOI: 10.1016/j.mcpro.2023.100630. Profiling of another drug that specifically targets a membrane protein that is not a kinase inhibitor could be informative of the general utility of this approach.

We agree with the reviewer that including drugs that targets membrane proteins would strengthen our manuscript. Therefore, we have performed two additional experiments, which have been included in the revised manuscript. These includes One-Tip-PISA experiments in both HeLa and SCC25 cells with treatment by 1 and 5 μ M of RMC4550, an allosteric inhibitor targeting the plasma-membrane bound phosphatase SHP2. Moreover, we also performed One-Tip-PISA experiments in SCC25 cells with a SERCA2 inhibitor (Thapsigargin) at 1 and 5 μ M as specifically suggested by the reviewer. These results are now displayed in Panel 6 as well as Supplementary Figure 12.

3. Figure 3A(1?) is not informative. If authors would like to include it should be in the supplement.

Figure 3 A has been moved to supplementary figures and the Figure 3 has generally been moved to the supplementary.

Overall, low input amount experiments with STS are not very novel (Fig. 3 & 4). Lots of PISA and CETSA experiments have been performed with STS and the low cell number data is likely to be noisier and there is no scientific justification for why such low cell amounts need to be used for benchmarking. This comes across as appealing to a trend rather than something with scientific merit.

More discussion and literature references have been added to emphasise the importance of discrete cell types which are limited in available cell number stressing the importance of sensitive sample preparation methods like One-Tip-PISA.

STS is so non-specific that it is easy to call any kinase a target. Overall data not informative in this reviewer's opinion. Who needs to know a lower input limit for easy to grow HeLa cells? Both Figure 3 & 4 are not of broad interest.

Figure Panel 2 has been moved to the supplementary Figure Panel 3 was expanded by a Volcano plot of SCC25 cells treated with a SHP-2inhibitor RMC4550. ANOVA table of Figure 4 has been moved to the supplementary. New Figure Pannel 6 was added to analyse One-Tip-PISA data collected on SCC25 cells.

4. Comparison of One Tip PISA to PISA-PAC is technically informative, but again provides few biological insights. Data is clear and highly correlated across conditions. Again, Figure 5(3) is not highly informative but does provide some technical insights and benchmarking giving more novel analytical insights than provided by Figures 3 & 4.

The ANOVA table from panel 3 has been placed in the supplementary figures.

5. Figure 6 has the most interesting data for this manuscript, and it has been heavily summarized. Drugs used are inhibitors of the EGFR pathway. The description of the results is fine, but discussion on why known target kinases are not identified as altered is needed.

Figure 5 (Previously figure 6) has been reworked. RMC4550 and Erlotinib have been added to the inhibitors used in the experiment. More discussion regarding targets and missing targets has been added including.

Additional Figure 6 has been created where more inhibitors have been tested on SCC25 cells including a SERCA2 inhibitor as suggested by the author.

Additional analysis showing if targets were not found to be stabilized/alterd because they were not detected should be included.

Additional discussion for missing kinase targets has been added into the manuscript.

Description of the number of membrane proteins and membrane associated kinases should be included.

In the new Figure 6 an analysis of membrane associated proteins as well as other subcellular locations has been performed.

These aspects of the study would be much more informative in the 96-well screen than in the earlier sections focused only on the highly studied STS. Suggest that the analysis and figures of this section be expanded to provide more molecular insights **including individual target measurement reproducibility across different treatment conditions**. Would be interesting to see kinase performance across the panel of inhibitors and the overlap of significant hits since there was pathway justification for the inhibitor selection.

Whilst we agree with the reviewer that even more biological insight would be of interest would we like to emphasise that the scope of the article is in the method development and proof of concept of a streamlined method to determine drug target complexes in a simple, elegant and non-automated way.

Minor comments:

1. Axis labels should include the conditions compared in the volcano plots for log₂ fold-change.

All axes of volcano plots have been adapted to show the treatment conditions.

2. Sentence: "The heatmap shows that across all subcellular fractions, ...c; lysosomal, vacuolar, and mitochondrial showed" should be -- endoplasmic reticulum, lysosome, vacuole, and mitochondria showed"

The sentence has been changed as suggested by the reviewer.

3. Figure 3 legend is labeled as Figure 1. Figure 4 legend is labeled as Figure 2. Figure 5 is labeled as Figure 3 – this is disruptive to reading the manuscript.

Conversion problem from word to PDF has been solved.

4. Data in Figure 6A are difficult to read because of the large number of colors used. Recommend reducing the color usage. Stars in Fig. 6D are nearly impossible to see by eye. Recommend using dots rather than stars.

Higher contrast has been added to the heatmaps in panel 5 and 6. Stars have been exchanged to large black dots for better readability.

Point-by-point rebuttal (COMMSCHEM-24-0630A).

“*High-throughput proteome integral solubility alteration assay for low cell input using One-Tip*” by Maico Lechner et al.

Our answers to the reviewer questions are indicated in **green text**

Reviewer #2 (Remarks to the Author):

Overall: This work establishes a method for streamlined sample preparation for PISA samples with post-heat treatment processing using a One-Tip method on an Evosep cartridge. This includes assessment of assay performance in a high throughput format, which will be highly useful for the larger drug-target interaction analysis research community. The authors have made many improvements in the manuscript and were very responsive to reviewers' comments. Overall, the manuscript is improved in many ways. Some points remain that should be addressed prior to publication, as discussed below.

Revision points to consider/address:

Figure 2 for the reviewers should be included in the manuscript. It is fine to include this as supplementary material. The advantages of different detergents and/or workflows for membrane protein recovery is of broad general interest. If the figure could be included as a main text figure, that could also be done but that should be up to the authors on where to place these comparisons in their narratives.

ANSWER: We thank the reviewer for this suggestion. The supplementary material have been updated accordingly with the Figure 2 for reviewers, which is now supplementary figures 5. We have also added a paragraph about membrane proteins identified in the three datasets in the main text of the article.

Figure 2B (table) needs higher contrast font for improved readability.

ANSWER: This has been corrected, the heatmap has now a higher contrast for increased readability.

Figure 3D: Spearman's correlation is written as speerman correlation. Since the test is a person's name, it should be capitalized in addition to the spelling correction. It is misspelled next to the color scale too.

ANSWER: This has been corrected

The addition of the RMC4550 treatments add value to the manuscript with the demonstration of a clear drug-target based stabilization. Volcano plots in Fig. 3E are also much clearly than those presented in Fig. 2C. I would suggest that the plot in 2C be reformatted to match 3E for ease of reading. Text on Figure 2C is also very small.

ANSWER: We thank the reviewer for highlighting the value of the RMC4550 experiments. We have reformatted Figure 2C as suggested, and kinases have been highlighted more clearly for better readability.

The rebuttal states that high CV levels are due to PISA sample handling without showing any evidence of that. It is still not clear that the high CVs are due to PISA steps rather than the One-Tip approach. This was not convincingly addressed. It is stated that the CV values differ for various drug treatment conditions in the large-scale screen and that it is “not ideal”. Why would there be different CV values across treatment conditions? Treatment conditions should not alter sample preparation reproducibility. This challenge will reduce the number of statistically significant hits.

ANSWER: The increased CV values observed in some treatment conditions are most likely due to experimental variation, such as pipetting errors. An illustrative example is shown in Figure 1, where we overlay the log₂-transformed intensities of the 1 μM AKT1/2 treatment in HeLa cells across four biological replicates. Notably, replicate 2 consistently exhibits higher signal intensities, which could likely be attributed to the accidental pipetting of a slightly larger sample volume. Since replicates are intended to mitigate the influence of such experimental errors, a median normalization across replicates should have been applied prior to calculating and plotting the CV values. We have now implemented this normalization step, and all relevant plots in the article have been updated accordingly. As a result, the overall CV values are markedly reduced, providing a more accurate representation of variability.

Figure 1: Histogram of log₂ normalized signal intensity of 1μM treatment of AKT1/2 inhibitor on HeLa cells with each replicate indicated by individual colors.

In Figure 2 and Figure 3, the impact of a median normalization between the 4 replicates of each treatment to eliminate the impact of experimental errors, which is common practice in proteomics data analysis. Experimental errors become evidently less impactful within the 4 replicates for each group when compared to Figure 3.

Figure 2: Box plots of non-median normalized CV values

Figure 3: Box plots of median normalized CV values

Please evaluate the manuscript again for typos. See particularly concerning examples such as “howeverwith TPP ewith TPP vwith TPP ewith TPP wwith TPP owith TPP this was not the case for our dataset “

ANSWER: We have done our best to go through and correct spelling mistakes.

Statement,

“Lastly Enzastaurin the PKC β inhibitor shows no significant stabilization of PKC β giving the same results as for treatment on HeLa cells thus implying that this specific drug target complex might not be detectable utilizing a PISA assay.”

Can this point be expanded? Was this typically shown by TPP or CETSA? Is it fair to say that this drug-target complex is not detectable or that the drug-target complex might not result in a significant change in target protein stability? It is unclear that this is specific to PISA and not the drug-target association. Earlier in the manuscript it is stated that, “PISA may not be able to solubilize the 375 inhibitor-target complex with PKC β , which could explain its absence in the dataset”. These two points are inconsistent with each other and need more clarity in both parts of the manuscript.

ANSWER: We thank the reviewer for this helpful comment. We agree that the original statements lacked clarity and could be interpreted as inconsistent. The manuscript has been revised to clarify that the absence of a detectable thermal shift for PKC β upon Enzastaurin treatment does not necessarily indicate a lack of binding. Rather, it likely reflects the limitations of thermal shift-based methods like PISA, which are known to produce false negatives, particularly for large, multi-domain proteins such as PKC β .

We now explicitly note that factors such as assay conditions and protein structure can impact the detectability of thermal shifts. In contrast to PKC β , the smaller, more compact GSK3B, which we detected as a target of Enzastaurin, may respond to ligand binding with more uniform stabilization. We also reference kinobead data from Reinecke et al., which showed higher affinity for GSK3B than PKC β , supporting this observation.

The revised text emphasizes that complementary approaches, such as phospho-proteomics or affinity pull-downs, are important for confirming drug target interactions, and that the lack of a PISA signal does not equate to a lack of binding. Both relevant sections of the manuscript have been updated accordingly.

Supplementary Table 1 and 2 – font is too small to be clearly readable.

ANSWER: The text in the tables and headings have been increased in size for better readability

Reviewer #1 (Remarks to the Author):

In this manuscript, the authors aim to develop the One-Tip-PISA method by combining PISA with the One-Tip technology to minimize sample handling and reduce the need for large input amounts for drug-target profiles. The authors first indicated the advantage of DDM over PBS in kinase detection, and evaluated the effects of varying DDM concentrations on cell lysis, ultimately selecting a 0.2% DDM concentration for their studies. The One-Tip-PISA method exhibited more potential than the traditional PISA-PAC approach especially for low cell input, of which sensitivity limit reached 200 cells/L. Furthermore, the authors adapted the One-Tip-PISA method to a 96-well plate format, utilizing a kinase inhibitor panel to profile both known targets and off-targets successfully. Overall, this novel integration of One-Tip-PISA is not only reproducible and efficient but also highly sensitive, allowing for an easy processing of intact cells in a 96 well format. Specific comments are listed as following:

We thank the reviewer 1 for his kind words

Major comments:

1. According to Reference 11, One-Tip technology has already been applied for single-cell proteomics, and showed high reproducibility and quantitative precision, with a remarkable coefficient of variance of 5–8% (3000 cells to 1000 cells). However, in this work, as shown in Figure 5A, the CV value was 30% for 2500 HeLa cells (200 cells/L) treated by DMSO (vehicle). What accounts for this kind of variance?

An explanation of why a higher variance in One-Tip-PISA is observed compared to only One-Tip sample preparation was added to the text. In short, this is inherent to PISA sample preparation which includes, low-efficiency cell lysis and many additional pipetting steps and sample handling which can lead to variation. For, instance, extra steps including, sample pooling, pipetting and centrifugation for supernatant removal thus increasing the overall variance.

2. In line 262, PAC detects significantly more kinases than One-Tip-PISA, particularly at 800 and 600 cells/L. It seemed that PAC was better for kinases identification if low cell input (200 cells/L) was not necessary. Therefore, it is essential to provide more **discussion** to emphasize the significance of low cell input in PISA, which will make it clear why such a threshold is necessary and how it enhances the applicability of One-Tip-PISA for limited cells.

More discussion and literature references have been added to emphasise the importance of discrete cell types which are limited in available cell number stressing the importance of sensitive sample preparation methods like One-Tip-PISA.

3. In Figure 5, when comparing these two methods (One-Tip-PISA and PISA-PAC), the number of identified proteins and peptides should be given for each treatment.

In Figure 4 (Previously Figure 5) the protein and peptide numbers have now been added in Figure 5C to compare PAC and One-Tip-PISA directly.

Minor points:

1. In line 87, I didn't find any discussion about 1%DDM from Reference 14, please check it.

The source has been changed, the relevant information can be found in the Supplementary Text and Figures of the article.

2. In Figure 3A, the deviation of 400 cells/L (10 μ M STS) was too large that the BCA assay might not be suitable for such a low concentration of cells treated by STS.

This is correct. The intention with this figure was to give researchers an idea about what protein identification numbers they can expect from different cell dilutions. This figure was moved to supplementary figures because it is of lower importance compared to other figures.

3. In line 263-265, 5 L of lysed cell was loaded for One-Tip-PISA while 25 ng of peptide was loaded for PAC-PISA. The sample loading was different? If the initial processing amount was equivalent, please make it clear.

An explanation to the different sample loads was added in the text to help clarify the differences in the sample loading of One-Tip-PISA (proteins are digested above the stationary phase of the tip) and PAC (Proteins are digested in a 96-well plate and peptides are loaded on EvoTips) and thus the variation and differences in detected kinases.

4. Please correct the number in figure legends of Figure 3~5.

The numbers of the figures in the PDF have been corrected

Reviewer #2 (Remarks to the Author):

This work establishes a method for streamlined sample preparation for PISA samples with post-heat treatment processing using a One-Tip method on an Evosep Evotip cartridge. The research team also evaluated input minimums for the PISA method for acquisition with DIA on an Orbitrap Astral. A limitation of this study is the focus on staurosporine-treated HeLa cells. From a benchmarking perspective, staurosporine has been used for multiple previous CETSA/PISA studies. The benefit to that is the ability to compare these datasets to many other datasets for comparison of proteome depth of coverage, number of positive target ids, and utility of the DIA method. The limitation is that staurosporine is an extremely broad, low specificity, kinase inhibitor. As such, there are numerous positive hits giving a possibly overly optimistic impression of assay sensitivity for this new method. If more selective inhibitors are used, the ability to use low input material for the PISA studies has not been established with this work. The positive aspects of this work are the integrated samples preparation method in a 96-well format, although that is not an entirely unique approach since it builds on a large foundation of previous work. The data from the 96-well inhibitor screen are interesting but are underdeveloped relative to other described findings that have lower overall novelty. The manuscript would benefit from editing to refocus on the 96-well screen with some of the low cell input datasets being removed or deemphasized.

We thank the reviewer for the feedback and constructive comments, and we agree that the focus of the article should be broadened.

Specific major comments:

1. It is stated that DDM enhances membrane protein recovery after PISA shown in Fig. 1B and C but should be referring to Figure 2B and C.

We thank the reviewer for pointing this out. It is correct that Figure 1A and B only show a difference in the number of identified proteins and peptides by using DDM instead of PBS. Therefore, it was not appropriate to refer to these Figures for the higher recovery of membrane proteins, which was introduced in Figure 2 B and C. Consequently, the text has now been changed to make this point more evident.

States a t-test was performed which is not the proper statistical test for protein class enrichment however the figure legend states that it was a 1-way ANOVA. This was a major point in the manuscript and should be clarified and clearly addressed to show the utility of DDM for PISA.

We are sorry about the confusion. The mentioned t-test was referring to figure 1B and C. Where a Welch's t-test was performed to compare the average identified proteins and peptides between different DDM concentrations and the PBS control. In Figure 2B, a 1-way ANOVA was performed and evaluated via a F-Test to determine if there are statistically significant differences in the different cell compartments while using DDM or PBS. The ANOVA results table was moved to the supplementary information and a more detailed explanation has been added to make it more clear to the reader which tests have been performed.

Comparisons to membrane protein yields in DOI: 10.1038/nmeth.3652 should be performed.

We thank the reviewer for this suggestion, which we have followed. The available tables from DOI: 10.1038/nmeth.3652 have been downloaded and compared to two datasets. The one produced in this paper using DDM and a PAC digest (Lechner et al) as well as with doi.org/10.1038/s41467-024-53240-2 (Bath et al) utilizing NP40 with a PAC digest. The results of the analysis can be found in the following plots.

Figure1: Bar chart of qualitative evaluation of identified membrane proteins recovered by using 0.2% NP40 Bath et.al, 0.2% DDM Lechner et.al and 0.4% NP40 Savitski et.al on HeLa cells and DMSO treatment. (Bath et.al and Lechner et.al use PAC digest since OneTip PISA can't be done with NP40 since it is not MS compatible)

Figure 1 compares qualitative number of membrane proteins in each data set. Indicating the using 0.2% DDM is comparable to 0.2% NP40. (Note Lechner et.al uses Orbitrap Astral MS, Bath et.al Orbitrap Exploris 480 and Savitski et.al Q-Exactive Orbitrap with TMT labelling and pre-run fractionation)

UpSet Plot of Proteins with cell membrane localization

Figure 2: Upset-Plot comparing membrane proteins in common and individual to each of the datasets

Figure 2 highlights that with 0.2% DDM and the PAC digest a substantial amount of membrane proteins can be recovered with the largest number of membrane proteins identified in this dataset with

210 whereas Bath et al finds 21 and Savitski et.al 26 specific membrane proteins. Largest overlap is also between Bath et al and Lechner et al 128 shared membrane proteins between the datasets.

Figure 3: Quantitative comparison of log₂ summed intensity of all membrane and non-membrane proteins displayed as stack plot between the 3 datasets.

Evaluating the difference between DDM and NP40 shows that there is a similar ratio between membrane and non-membrane proteins while using PAC Bath et al 96.7% to 3.3% and Lechner et al 95.3% to 4.7%. This however is not true for the dataset from Savitski et.al where the ratio is 99.9% to 0.1%

Comparisons to PBS rather than another non-ionic detergent are not informative. Suggest comparison to 0.4% NP40 based on the published work above, although the comparison to PBS can still be included and could be informative to some research groups.

We agree with this statement, however, the purpose of DDM in One-Tip-PISA is a necessity to streamline the method and increase sample throughput, as it is a MS-friendly detergent that is directly compatible with LC-MS. That is not the case for NP40, which needs to be removed prior to MS analysis, f.ex. through a PAC digestion workflow. A comparison between DDM and NP40 on HeLa cells and a PAC digest is displayed on Figures 1-3 above.

It is expected that additional processing steps for samples extracted with 0.4% NP40 would decrease yields but this is not shown.

We agree with the reviewer's statement, but a fair comparison between detergents is not possible as very low sample input with a PAC digest will not work. The scope of the article is not to introduce DDM as a detergent or compare it against other detergents it is a necessary part of the sample preparation to use the One-Tip-PISA method as a MS compatible detergent is mandatory.

Fig. 2D is not convincing to this reviewer that the effect is specific to 0.2% DDM, could also be a replicate effect.

Table 1: Grubbs Test results testing for outliers in 10µM STS in PBS and DMSO control in PBS

Condition	Values	Mean	Standard Deviation	Grubbs' Statistic	Critical Value	Outlier Present
10µM STS in PBS	[5511, 5571, 5318, 5629]	5507.3	135.1	1.401	1.481	FALSE
DMSO in PBS	[5232, 5468, 5084, 5193]	5244.3	161.8	1.383	1.481	FALSE

In Table 1 we tested both datasets 10µM STS in PBS and DMSO control in PBS for outliers to make sure the change in variance is not connected to a replica effect during the experiment. In both cases the Grubbs-test resulted in accepting H_0 that there are no outliers in the dataset.

Figure 4: CV values for each dataset comparing the effects of different DDM concentrations against a PBS control.

Considering Figure 4 we can see that both DMSO PBS, and 10 μ M STS PBS show a larger median CV value than most of the DDM containing conditions thus indicating that the skewed distribution of the volcano plot is a result of the lack of DDM instead of just a replicate issue.

The shape of the volcano plot overall between PBS and DDM is very different, even though the scale is identical. What is the reason for that difference?

-I would argue that this is due to the effect of DDM on stabilising proteins and making sure that they e.g. do not get stuck so much at the surface of a tube. Hence the lower variance in the DDM fraction.

0.2% DDM vs PBS log₂ intensity as distribution of different categories (Histogram) then we see two distributions with and without DDM to convince that there is a quantitative difference

Figure 5: Histograms displaying the log₂ signal intensity against the frequency comparing 10 μ M STS lysed with 0.2% DDM against PBS (left) and DMSO control lysed with 0.2% DDM against PBS (right)

As seen in the article Figure 1B and C a lack of DDM leads to lower overall protein and peptide IDs. Moreover, in Figure 5 the log₂ intensity between PBS and DDM lysed samples show that there is a strong quantitative difference between the two conditions. Samples that have only been lysed with PBS do not result in high signal intensity at a high frequency as often as while using DDM. This implies a lower certainty of quantification and higher variability which is one of the reasons why the volcano plot is skewed as the quantitation accuracy is lower. Furthermore, DDM is known to improve the recovery of hydrophobic proteins by stabilizing them which would be lost in the PBS fraction, thus leading to a larger overall variance in the PBS datasets compared to the ones where DDM is used.

Fig. 2E data and could be moved from Supp. Figure 2 (although log₂ ratio should be defined, EGFR looks destabilized by STS based on the figure title but in text it is stated it is stabilized).

Supplementary figure 2 has been moved from the supplementary to the main panel 2 and the text has been clarified.

Correlation values in Supp Fig 4 are quite convincing of DDS extraction reproducibility. Comparison to 0.4% NP40 would be highly informative. It is no surprise based on prior work that nonionic detergent is beneficial over PBS.

A comparison between DDM and NP40 has been implemented earlier during the revision in Figure 1-3.

2. Membrane protein SERCA2 as a target ID was shown in DOI: 10.1016/j.mcpro.2023.100630. Profiling of another drug that specifically targets a membrane protein that is not a kinase inhibitor could be informative of the general utility of this approach.

We agree with the reviewer that including drugs that targets membrane proteins would strengthen our manuscript. Therefore, we have performed two additional experiments, which have been included in the revised manuscript. These includes One-Tip-PISA experiments in both HeLa and SCC25 cells with treatment by 1 and 5 μ M of RMC4550, an allosteric inhibitor targeting the plasma-membrane bound phosphatase SHP2. Moreover, we also performed One-Tip-PISA experiments in SCC25 cells with a SERCA2 inhibitor (Thapsigargin) at 1 and 5 μ M as specifically suggested by the reviewer. These results are now displayed in Panel 6 as well as Supplementary Figure 12.

3. Figure 3A(1?) is not informative. If authors would like to include it should be in the supplement.

Figure 3 A has been moved to supplementary figures and the Figure 3 has generally been moved to the supplementary.

Overall, low input amount experiments with STS are not very novel (Fig. 3 & 4). Lots of PISA and CETSA experiments have been performed with STS and the low cell number data is likely to be noisier and there is no scientific justification for why such low cell amounts need to be used for benchmarking. This comes across as appealing to a trend rather than something with scientific merit.

More discussion and literature references have been added to emphasise the importance of discrete cell types which are limited in available cell number stressing the importance of sensitive sample preparation methods like One-Tip-PISA.

STS is so non-specific that it is easy to call any kinase a target. Overall data not informative in this reviewer's opinion. Who needs to know a lower input limit for easy to grow HeLa cells? Both Figure 3 & 4 are not of broad interest.

Figure Panel 2 has been moved to the supplementary Figure Panel 3 was expanded by a Volcano plot of SCC25 cells treated with a SHP-2inhibitor RMC4550. ANOVA table of Figure 4 has been moved to the supplementary. New Figure Pannel 6 was added to analyse One-Tip-PISA data collected on SCC25 cells.

4. Comparison of One Tip PISA to PISA-PAC is technically informative, but again provides few biological insights. Data is clear and highly correlated across conditions. Again, Figure 5(3) is not highly informative but does provide some technical insights and benchmarking giving more novel analytical insights than provided by Figures 3 & 4.

The ANOVA table from panel 3 has been placed in the supplementary figures.

5. Figure 6 has the most interesting data for this manuscript, and it has been heavily summarized. Drugs used are inhibitors of the EGFR pathway. The description of the results is fine, but discussion on why known target kinases are not identified as altered is needed.

Figure 5 (Previously figure 6) has been reworked. RMC4550 and Erlotinib have been added to the inhibitors used in the experiment. More discussion regarding targets and missing targets has been added including.

Additional Figure 6 has been created where more inhibitors have been tested on SCC25 cells including a SERCA2 inhibitor as suggested by the author.

Additional analysis showing if targets were not found to be stabilized/alterd because they were not detected should be included.

Additional discussion for missing kinase targets has been added into the manuscript.

Description of the number of membrane proteins and membrane associated kinases should be included.

In the new Figure 6 an analysis of membrane associated proteins as well as other subcellular locations has been performed.

These aspects of the study would be much more informative in the 96-well screen than in the earlier sections focused only on the highly studied STS. Suggest that the analysis and figures of this section be expanded to provide more molecular insights **including individual target measurement reproducibility across different treatment conditions**. Would be interesting to see kinase performance across the panel of inhibitors and the overlap of significant hits since there was pathway justification for the inhibitor selection.

Whilst we agree with the reviewer that even more biological insight would be of interest would we like to emphasise that the scope of the article is in the method development and proof of concept of a streamlined method to determine drug target complexes in a simple, elegant and non-automated way.

Minor comments:

1. Axis labels should include the conditions compared in the volcano plots for log₂ fold-change.

All axes of volcano plots have been adapted to show the treatment conditions.

2. Sentence: "The heatmap shows that across all subcellular fractions, ...c; lysosomal, vacuolar, and mitochondrial showed" should be -- endoplasmic reticulum, lysosome, vacuole, and mitochondria showed"

The sentence has been changed as suggested by the reviewer.

3. Figure 3 legend is labeled as Figure 1. Figure 4 legend is labeled as Figure 2. Figure 5 is labeled as Figure 3 – this is disruptive to reading the manuscript.

Conversion problem from word to PDF has been solved.

4. Data in Figure 6A are difficult to read because of the large number of colors used. Recommend reducing the color usage. Stars in Fig. 6D are nearly impossible to see by eye. Recommend using dots rather than stars.

Higher contrast has been added to the heatmaps in panel 5 and 6. Stars have been exchanged to large black dots for better readability.

Second Revision

Reviewer #2 (Remarks to the Author):

Overall: This work establishes a method for streamlined sample preparation for PISA samples with post-heat treatment processing using a One-Tip method on an Evosep cartridge. This includes assessment of assay performance in a high throughput format, which will be highly useful for the larger drug-target interaction analysis research community. The authors have made many improvements in the manuscript and were very responsive to reviewers' comments. Overall, the manuscript is improved in many ways. Some points remain that should be addressed prior to publication, as discussed below.

Revision points to consider/address:

Figure 2 for the reviewers should be included in the manuscript. It is fine to include this as supplementary material. The advantages of different detergents and/or workflows for membrane protein recovery is of broad general interest. If the figure could be included as a main text figure, that could also be done but that should be up to the authors on where to place these comparisons in their narratives.

The figures from the last rebuttal letter have been updated, placed in the supplementary as supplementary figures 5 and a paragraph about membrane proteins of the three datasets has been implemented in the main text of the article.

Figure 2B (table) needs higher contrast font for improved readability.

This has been corrected, the heatmap has now a higher contrast for increased readability.

Figure 3D: Spearman's correlation is written as speerman correlation. Since the test is a person's name, it should be capitalized in addition to the spelling correction. It is misspelled next to the color scale too.

This has been corrected

The addition of the RMC4550 treatments add value to the manuscript with the demonstration of a clear drug-target based stabilization. Volcano plots in Fig. 3E are also much clearly than those presented in Fig. 2C. I would suggest that the plot in 2C be reformatted to match 3E for ease of reading. Text on Figure 2C is also very small.

Pannel 2 and in particular Figure 2C have been reformatted, and kinases have been highlighted more clearly for better readability.

The rebuttal states that high CV levels are due to PISA sample handling without showing any evidence of that. It is still not clear that the high CVs are due to PISA steps rather than the One-Tip approach. This was not convincingly addressed. It is stated that the CV values differ for various drug treatment conditions in the large-scale screen and that it is “not ideal”. Why would there be different CV values across treatment conditions? Treatment conditions should not alter sample preparation reproducibility. This challenge will reduce the number of statistically significant hits.

The increased CV values observed in some treatment conditions are most likely due to experimental variation, such as pipetting errors. An illustrative example is shown in Figure 1, where we overlay the log₂ intensities of the 1 μM AKT1/2 treatment in HeLa cells across four biological replicates. Notably, replicate 2 consistently exhibits higher signal intensities, which could be attributed to the accidental pipetting of a larger sample volume. Since replicates are intended to mitigate the influence of such experimental errors, a median normalization across replicates should have been applied prior to calculating and plotting the CV values. We have now implemented this normalization step, and all relevant plots in the article have been updated accordingly. As a result, the overall CV values are markedly reduced, providing a more accurate representation of variability.

Figure 1: Histogram of log₂ normalized signal intensity of 1 μM treatment of AKT1/2 inhibitor on HeLa cells with each replicate indicated by individual colors.

In Figure 2 and Figure 3, the impact of a median normalization between the 4 replicates of each treatment to eliminate the impact of experimental errors, which is common practice in proteomics data analysis. Experimental errors become evidently less impactful within the 4 replicates for each group when compared to Figure 3.

Figure 2: Box plots of non-median normalized CV values

Figure 3: Box plots of median normalized CV values

Please evaluate the manuscript again for typos. See particularly concerning examples such as “howeverwith TPP ewith TPP vwith TPP ewith TPP wwith TPP owith TPP this was not the case for our dataset “

Spelling mistakes have been corrected

Statement,

“Lastly Enzastaurin the PKC β inhibitor shows no significant stabilization of PKC β giving the same results as for treatment on HeLa cells thus implying that this specific drug target complex might not be detectable utilizing a PISA assay.”

Can this point be expanded? Was this typically shown by TPP or CETSA? Is it fair to say that this drug-target complex is not detectable or that the drug-target complex might not result in a significant change in target protein stability? It is unclear that this is specific to PISA and not the drug-target association. Earlier in the manuscript it is stated that, “PISA may not be able to solubilize the 375 inhibitor-target complex with PKC β , which could explain its absence in the dataset”. These two points are inconsistent with each other and need more clarity in both parts of the manuscript.

We thank the reviewer for this helpful comment. We agree that the original statements lacked clarity and could be interpreted as inconsistent. The manuscript has been revised to clarify that the absence of a detectable thermal shift for PKC β upon Enzastaurin treatment does not necessarily indicate a lack of binding. Rather, it likely reflects the limitations of thermal shift-based methods like PISA, which are known to produce false negatives, particularly for large, multi-domain proteins such as PKC β .

We now explicitly note that factors such as assay conditions and protein structure can impact the detectability of thermal shifts. In contrast to PKC β , the smaller, more compact GSK3B, which we detect as a target, may respond to ligand binding with more uniform stabilization. We also reference kinobead data from Reinecke et al., which shows moderate binding selectivity of Enzastaurin to GSK3B, supporting this observation.

The revised text emphasizes that complementary approaches, such as phospho-proteomics or affinity pull-downs, are important for confirming drug target interactions, and that the lack of a PISA signal does not equate to a lack of binding. Both relevant sections of the manuscript have been updated accordingly.

Supplementary Table 1 and 2 – font is too small to be clearly readable.

The text in the tables and headings have been increased in size for better readability